# ExPLAIND: Unifying Model, Data, and Training Attribution to Study Model Behavior

**Florian Eichin** [1 2]  **Yupei Du** [3]  **Philipp Mondorf** [1 2]  **Maria Matveev** [2 4]  **Barbara Plank** [1 2]  **Michael A. Hedderich** [1 2]

## Abstract

Post-hoc interpretability methods typically attribute a model's behavior to its components, data, or training trajectory in isolation, and are often tied to a particular level of granularity along the local-to-global spectrum. This leads to explanations that lack a unified view and may miss key interactions. We present ExPLAIND, a theoretically grounded, unified framework that integrates model components, data, and training trajectory while supporting explanations across granularities. We generalize recent work on gradient path kernels, reformulating models trained by AdamW as kernel machines. From the resulting kernel feature maps, we derive novel parameter-wise and step-wise influence scores. We empirically validate the resulting decomposition of model behavior in several settings and apply ExPLAIND to two case studies. Our findings on a Transformer exhibiting Grokking support previously proposed learning phases, while refining the final phase as one in which outer layers align around a representation pipeline learned after memorization. For EuroLLM pretraining, ExPLAIND reveals a two-phase dynamic, with the first characterized by outer-layer MLP learning and the second by increased relative influence of intermediate attention layers. These results establish ExPLAIND as a unified framework for interpreting model behavior and training dynamics.

---

[1]MaiNLP, Center for Information and Language Processing, LMU Munich, Germany [2]Munich Center for Machine Learning (MCML) [3]Saarland Informatics Campus, Saarland University, Germany [4]Department of Mathematics, LMU Munich, Germany. Correspondence to: Florian Eichin <feichin@cis.lmu.de>.

*Proceedings of the $43^{rd}$ International Conference on Machine Learning*, Seoul, South Korea. PMLR 306, 2026. Copyright 2026 by the author(s).

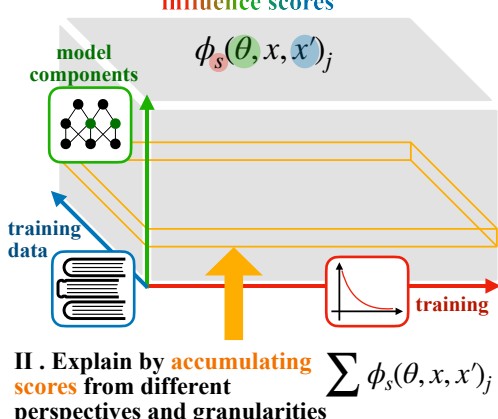

**I. Model decomposes into influence scores**

model components

$\phi_s(\theta, x, x')_j$

training data

training

**II. Explain by accumulating scores from different perspectives and granularities**

$$\sum \phi_s(\theta, x, x')_j$$

*Figure 1.* The ExPLAIND framework is based on a decomposition of the model along its components, data, and training steps. Explanations are obtained by accumulating the resulting scores.

## 1. Introduction

Understanding the latent mechanisms of deep neural networks remains one of the central challenges in machine learning (Rudin et al., 2021; Rai et al., 2024; Zhang et al., 2025). As models grow in complexity, interpretability has become important not only for debugging and improving models, but also for helping humans scrutinize model behavior in light of concerns such as fairness, safety, and trust (Doshi-Velez & Kim, 2017). Much of the recent progress in interpretability has focused on attributing a model's behavior to one of three main factors: its components, the data it was trained on, or the dynamics of the training itself. At the same time, interpretability methods are often tied to a particular level of granularity along the local-to-global spectrum, yielding either fine-grained, local explanations or broader, global characterizations.

However, these approaches are often applied in isolation across both perspective and granularity. Explanations focused on model components may ignore the influence of individual training examples or how these components evolved during optimization. Data-centric explanations can overlook how different parts of the model internalize those examples. Likewise, analyses at one level of granularity may

miss patterns that only emerge at another. This fragmentation limits our understanding, leaving important interactions unexplored. While some work has probed training dynamics (Müller-Eberstein et al., 2023; Tigges et al., 2024), their insights are only loosely connected to the model internals or data and lack a theoretical connection between checkpoints.

To address this fragmentation, we propose **Ex**act **P**ath-**L**evel **A**ttribution **I**ntegrating **N**etwork and **D**ata (**ExPLAIND**), a unified framework that captures how data, model components, and training dynamics jointly influence model behavior. ExPLAIND builds on the Exact Path Kernel (EPK, Bell et al., 2023) view of gradient-based learning, but extends it to modern training regimes. With this, ExPLAIND provides a theoretically grounded lens through which we can analyze how individual training examples influence model components throughout training. This unified perspective helps, for example, to interpret emergent learning phenomena such as Grokking (Power et al., 2022) or LLM training. Our work makes the following contributions:

(i) **Theoretical extension of the EPK.** We generalize the EPK to modern training regimes that include first- and second-order gradient estimates, weight decay, dynamic learning rates, and mini-batching (see Section 3). We empirically validate that it accurately represents CNNs on vision tasks, a Transformer on a math task, and even a 1.7B LLM trained on natural language.

(ii) **ExPLAIND framework.** We derive novel influence scores that quantify how individual parameters, training samples and training steps contribute to model predictions (see Section 4). The framework can be applied at different levels of granularity and from different perspectives, such as parameter level, data level and training-step level, cf. Figure 1. We validate the effectiveness of these scores for model component attribution via parameter pruning.

(iii) **Case studies of Grokking and LLM pretraining.** To demonstrate the capabilities of ExPLAIND, we apply it in two case studies. For a widely studied modulo addition Transformer exhibiting Grokking, we uncover a previously unreported alignment phase where input embeddings and final layers align around a representation pipeline learned in the preceding training steps. Further, we study the pretraining checkpoints of EuroLLM (Martins et al., 2024), identifying two main learning phases, of which the first is characterized by outer-layer MLP learning and the second by an increased relative influence of intermediate attention layers.

Thereby, ExPLAIND offers a theoretically grounded, empirically validated, and practically useful framework for analyzing machine learning models in a holistic manner.

## 2. Related work

We highlight the directly relevant works and provide a more detailed discussion of the related literature in Appendix D.

Post-hoc interpretability methods typically attribute model behavior to one of *input features* (Ribeiro et al., 2016), *training data* (Koh & Liang, 2017), or *model components* (Alain & Bengio, 2017). However, many approaches lack a theoretical foundation (Lipton, 2017; Saphra & Wiegreffe, 2024; Doshi-Velez & Kim, 2017; Basu et al., 2020; Bae et al., 2022) and can trade faithfulness for plausibility of explanations (Jacovi & Goldberg, 2020). Yolcu et al. (2025) propose the DualXDA framework, which combines novel, highly efficient data attribution with input feature attribution to provide explanations unified along data and input features. ExPLAIND further attributes the model parameters and training steps, but does not provide perspectives on the role of the input features.

Other work has extended interpretability into the temporal dimension, attributing model behavior to the *training dynamics*. For example, probing or circuit finding has been applied at different model checkpoints to identify learning phases (Müller-Eberstein et al., 2023; Tigges et al., 2024; Prakash et al., 2024). However, these approaches typically treat each training step independently, lacking a theoretical framework for *integrating changes in model behavior over time*. Loss Change Allocation (LCA) (Lan et al., 2019) and its recent extension POLCA (Kangaslahti et al., 2025) are similar to our approach, but focus on approximate solutions to decomposing the loss and lack the connection of the training data to the predictions.

Gradient path kernels (Domingos, 2020) reformulate a model trained by gradient descent as a kernel machine. Bell et al. (2023) extended this perspective to an exact equivalence, deriving the *Exact Path Kernel* (EPK) and studying data attribution through the kernel scores. Our notion of the tensor of influences generalizes this idea to the model components and training steps. Furthermore, their formulation does not cover realistic learning scenarios involving gradient updates based on first- and second-order estimates, weight decay, dynamic learning rates, and mini-batching. Central to the EPK reformulation is the stepwise comparison of training and test sample gradients via dot products. This connects the EPK to the Neural Tangent Kernel (Jacot et al., 2018), which it generalizes over the training trajectory. TracIn (Pruthi et al., 2020) also measures dot-products of gradients across training steps for data attribution, but lacks a theoretical connection to model predictions.

## 3. Exact Path Kernel Equivalence for AdamW

ExPLAIND is based on the EPK by decomposing the model predictions into fine-grained units of influence along the

training data, model parameters, and training steps. To capture the dynamics of modern optimization, including weight decay, moment estimates, learning rate schedules, and mini-batching, we focus on the AdamW optimizer (Loshchilov & Hutter, 2019) and the special case of gradient descent with momentum. Our derivation expresses the prediction change at each optimization step directly in terms of the parameter-gradients induced by the training samples, together with an additional term capturing the effect of decoupled weight decay. This yields an exact decomposition of the final prediction along these dimensions.

**Theorem 3.1** (Extension of Bell et al. (2023) to AdamW).
*Let $f_\theta : \mathcal{X} \to \mathcal{Y}$ be a model with parameters $\theta \in \mathbb{R}^D$, where $\mathcal{X} \subseteq \mathbb{R}^I$ and $\mathcal{Y} \subseteq \mathbb{R}^O$. Let $\mathcal{D} = \{(x_1, y_1), \ldots, (x_M, y_M)\}$ be a dataset, let $L : \mathcal{Y} \times \mathcal{Y} \to \mathbb{R}_{\geq 0}$ be a per-sample loss, and define the batch loss at training step $i$ by $\mathcal{L}_i(\theta) := \sum_{k \in B_i} w_{i,k} L(f_\theta(x_k), y_k)$, for batch $B_i \subseteq \{1, \ldots, M\}$ and $w_{i,k} \in \mathbb{R}$ the weight of sample $(x_k, y_k)$. Suppose that $\theta_N$ is obtained from $\theta_0$ by $N$ steps of AdamW with learning rates $\alpha_s \in \mathbb{R}_{>0}$, weight decay $\lambda \in \mathbb{R}_{\geq 0}$, betas $\beta_{1,2} \in [0, 1]$, bias-corrected second-moment terms $\hat{v}_{s+1}$, and stability constant $\epsilon \in \mathbb{R}_{>0}$ at step $s$. Then, for every $x \in \mathcal{X}$, the model prediction decomposes as follows*

$$
\begin{aligned}
f_{\theta_N}(x) = f_{\theta_0}(x) &- \sum_{k=1}^{M} \sum_{s=0}^{N-1} \phi_s^{test}(x)\, \phi_s^{train}(x_k) \\
&- \sum_{s=0}^{N-1} \phi_s^{test}(x)\, \mathbf{r}_s,
\end{aligned}
\tag{1}
$$

*where*

$$
\phi_s^{test}(x) := \int_0^1 \nabla_\theta f_{\theta_s(t)}(x)\, dt \in \mathbb{R}^{O \times D},
$$

$$
\phi_s^{train}(x_k) := \sum_{i=0}^{s} \alpha_{s,i}\, \mathbf{1}_{k \in B_i}\, w_{i,k}\, \frac{\nabla_\theta \big( L(f_{\theta_i}(x_k), y_k) \big)}{\sqrt{\hat{v}_{s+1}} + \epsilon}
$$
$$
\in \mathbb{R}^D,
$$

$$
\mathbf{r}_s := \alpha_s \lambda \theta_s \in \mathbb{R}^D,
$$
$$
\theta_s(t) := \theta_s + t(\theta_{s+1} - \theta_s) \in \mathbb{R}^D,
$$
$$
\alpha_{s,i} := \alpha_s (1 - \beta_1) \beta_1^{s-i} (1 - \beta_1^{s+1})^{-1} \in \mathbb{R},
$$

*Sketch of Proof.* We follow similar arguments as Bell et al. (2023). The key difference is to start from the AdamW parameter update and to account for mini-batches through indicator variables. The extended Theorem and complete proof is given in Appendix F.1. □

A formulation like the one in (Bell et al., 2023) with separate loss gradients follows directly (see Appendix F.1). Gradient descent with momentum decomposes analogously.[1]

---

[1]In fact, any regime descending along moving averages of the gradients as remarked by Bell et al. (2023)

**Corollary 3.2** (Gradient Descent with Momentum). *With setup as in Theorem 3.1, for gradient descent with momentum $\beta \in [0, 1]$, the same decomposition applies but with*

$$
\begin{aligned}
\phi_s^{train}(x_k) := \sum_{i=0}^{s} \big( &\alpha_s \beta^{s-i} \mathbf{1}_{k \in B_i}\, w_{i,k} \\
&\nabla_\theta L(f_{\theta_i}(x_k), y_k) \big) \in \mathbb{R}^D,
\end{aligned}
$$

$$
\mathbf{r}_s := \alpha_s \sum_{i=0}^{s} \beta^{s-i} \lambda \theta_i \in \mathbb{R}^D,
$$

*and $\phi_s^{test}(x)$ is as in Theorem 3.1.*

*Proof.* We follow a similar approach as in Theorem 3.1, substituting the relevant parts. The full proof can be found in Appendix F.1. □

We further derive the following corollary which decomposes the trajectories of intermediate model activations (such as layers) as well as differentiable functions of the output such as the loss, thus extending our framework to decomposing many other quantities used to study model behavior.

**Corollary 3.3** (Loss, activations, functions of output). *Assume the setup of Theorem 3.1 holds. Furthermore, let the model be of the form $f_\theta = h_\kappa \circ g_\iota$. Analogously to the decomposition in Theorem 3.1, we can decompose (i) the composition $h \circ f_{\theta_N}$ of the model output with any suitable differentiable function $h$, including in particular the loss $L(f_{\theta_N}(x), y)$, and (ii) the intermediate activations $g_\iota(x)$.*

*Proof.* For (i), we set $\tilde{f}_\theta(x) = h(f_\theta(x))$ and substitute it into the derivation of the test feature map and relevant substitutions. The loss directly follows as the special case $h_y(z) = L(z, y)$. For (ii), we set $\tilde{f}_\theta(x) = g_\iota(x)$ and apply Theorem 3.1 analogously. See Appendix F.3 for details. □

### 3.1. Validation of exact model representation

*Table 1.* We report the accuracy of the EPK representation with respect to model predictions, and the output similarity measured by KL divergence. For 100 integration steps, the EPK matches the trained model in all settings.

| Model | **ResNet9** | | **Transformer** | | **CNN** | |
|---|---|---|---|---|---|---|
| Data | CIFAR-2 | | MOD-113 | | MNIST | |
| Integr. Steps | 10 | 100 | 10 | 100 | 10 | 100 |
| EPK Acc. | 0.997 | 1.0 | 0.748 | 1.0 | 0.998 | 1.0 |
| KL Diverg. | 0.0 | 0.0 | 0.885 | 0.0 | 0.0 | 0.0 |

To empirically validate the exact equivalence of Theorem 3.1 and Theorem 3.2, we compute the EPK for a Transformer model, a standard CNN, and ResNet-9, trained on modulo addition, MNIST, and CIFAR-2 respectively.[2] For this, we

---

[2]Code at https://github.com/mainlp/explaind, details in Appendix E.

evaluate different numbers of integration steps in the test feature map $\phi^{\text{test}}$ and summarize the results in Table 1. The derived formulation provides an accurate approximation at 100 integration steps for all models, reproducing 100% of their classification decisions. Furthermore, the output distributions closely match those of the original models, indicated by near-zero KL divergences. We therefore use 100 integration steps for $\phi^{\text{test}}$ for the rest of this study.

## 4. The ExPLAIND framework

We now build on the established EPK of the training with realistic optimizers such as AdamW and define the ExPLAIND framework, which attributes prediction at the level of training samples, model parameters, and training steps. The key idea is a *tensor of influences* which indexes contributions by training steps, parameters, training samples, and outputs. Different explanations are then obtained by accumulating this tensor along selected axes, leading to parameter-, data-, and step-wise attributions at arbitrary levels of granularity, as visualized in Figure 1. We denote the influence of a parameter $\theta^{(i)}$ at step $s$ due to training sample $x_k$ on the prediction of output $j$ of a sample $x$ as

$$\psi(s, \theta^{(i)}, x_k, x)_j := \phi_s^{test}(x)_{j,i} \cdot \phi_s^{train}(x_k)_i$$

and the influence of the regularization on $x$ at step $s$ as

$$\psi^{reg}(s, \theta^{(i)}, x)_j := \phi_s^{test}(x)_{j,i} \cdot (\mathbf{r}_s)_i.$$

Rewriting Theorem 3.1 using this definition, the $j$-th coordinate of the model output can be written as

$$f_{\theta_N}(x)_j = f_{\theta_0}(x)_j - \sum_{k=1}^{M} \sum_{s=0}^{N-1} \sum_{i=1}^{D} \psi(s, \theta^{(i)}, x_k, x)_j$$
$$- \sum_{s=0}^{N-1} \sum_{i=1}^{D} \psi^{reg}(s, \theta^{(i)}, x)_j. \tag{2}$$

Hence, the scores $\psi(s, \theta^{(i)}, x_k, x)_j$ and $\psi^{reg}(s, \theta^{(i)}, x)_j$ are additive and sum up to the model prediction. The naturally arising backbone of our ExPLAIND framework is the accumulation of these scores over different (sub-)sets of parameters, training and test samples, as well as training steps. We make this explicit with the following definition. We collect all atomic scores into one multi-dimensional object, with axes corresponding to *steps*, *parameters*, *test samples*, *training samples*, and *outputs*. For a set of training steps $\mathcal{S} \subset \{1, 2, ..., N\}$, model parameters $\Theta \subset \{\theta^{(1)}, \theta^{(2)}..., \theta^{(D)}\}$, predicted samples $\mathcal{X}_{pred} \subset \mathcal{D}_{test}$, training samples $\mathcal{X} \subset \mathcal{D}_{train}$, and outputs $\mathcal{J} \subset \{1, ..., O\}$ we define the respective **tensor of influences** as

$$\Gamma(\mathcal{S}, \Theta, \mathcal{X}, \mathcal{X}_{pred})_{\mathcal{J}} := (\psi(s, \theta, x, x')_j)_{s, \theta, x, x', j}$$

where $s \in \mathcal{S}, \theta \in \Theta, x \in \mathcal{X}, x' \in \mathcal{X}_{pred}, j \in \mathcal{J}$ and, by convention, if $e$ is not already a set we identify it with the singleton $\{e\}$. Further, we define the **accumulated influence** as sums over selected axes of the tensor of influences:

$$\Psi(\mathcal{S}, \Theta, \mathcal{X}, \mathcal{X}_{pred})_{\mathcal{J}} := \text{sum}(\Gamma(\mathcal{S}, \Theta, \mathcal{X}, \mathcal{X}_{pred})_{\mathcal{J}})$$

where $\text{sum}$ sums up the elements along all dimensions of the tensor. Furthermore, we define leaving out a category in the arguments of $\Psi$ or $\Gamma$ as summing over the complete set along it. For example, $\Psi(\Theta, \mathcal{X}, \mathcal{X}_{pred})$ is the influence summed over all steps $\mathcal{S}$ and all output dimensions $\mathcal{J}$. To account for differences in sign across influences to different predictions and outputs, we further define the **accumulated importance** as follows

$$\bar{\Psi}(\mathcal{S}, \Theta, \mathcal{X}, \mathcal{X}_{pred})_{\mathcal{J}} := \sum_{x \in \mathcal{X}_{pred}} \sum_{j \in \mathcal{J}} |\Psi(\mathcal{S}, \Theta, \mathcal{X}, x)_j|$$

We denote the analogous definitions for the regularization scores with $\Gamma^{reg}$, $\Psi^{reg}$, and $\bar{\Psi}^{reg}$.

This formulation allows us to explain model behavior in a unified way across different dimensions. By choosing which axes of the influence tensor to keep and which to sum over, we obtain different perspectives such as parameter-level, data-level, or step-level attribution. Within each perspective, the *granularity* of the explanation depends on the size of the sets we consider. For example, if we want to study the influence of a single parameter $\theta^{(i)}$ on a given prediction of a sample $x$, we can accumulate parameter-wise kernel scores over the training set as $\Psi(\theta^{(i)}, x)$. In order to zoom out to the layer level per training step, we sum up the scores of all parameters that are a part of a layer $\Theta_L$ of the dataset at step $s$, i.e. compute $\Psi_s(\Theta_L, x)$. We provide more details and more examples in Appendix F.5 and illustrate the practical use of such explanations in Sections 5 and 6.

To investigate the relative influence of the regularization term, with analogous definitions and conventions as above, we furthermore define the **parameter-wise relative regularization influence** on the model behavior as the difference

$$D_{\mathcal{S}, \mathcal{X}, \mathcal{X}_{pred}, \mathcal{J}}(\Theta) := \sum_{\theta \in \Theta} (\Psi(\mathcal{S}, \theta, \mathcal{X}, \mathcal{X}_{pred})_{\mathcal{J}}$$
$$- \Psi^{reg}(\mathcal{S}, \theta, \mathcal{X}, \mathcal{X}_{pred})_{\mathcal{J}}).$$

When interpreting the topology of high dimensional spaces, studying the similarity of their elements can be a helpful tool. With analogous definitions and conventions as above, we define the **similarity of two predictions** $x, x' \in \mathcal{X}$ in terms of the cosine similarity of their respective scores:

$$Sim_{\mathcal{S}, \Theta, \mathcal{X}, \mathcal{J}}(x, x') :=$$
$$\frac{\text{flat}(\Gamma(\mathcal{S}, \Theta, \mathcal{X}, x)_{\mathcal{J}}) \cdot \text{flat}(\Gamma(\mathcal{S}, \Theta, \mathcal{X}, x')_{\mathcal{J}})}{||\Gamma(\mathcal{S}, \Theta, \mathcal{X}, x)_{\mathcal{J}}|| \cdot ||\Gamma(\mathcal{S}, \Theta, \mathcal{X}, x')_{\mathcal{J}}||}$$

where flat transforms a tensor into a vector and $||\cdot||$ is the vector $\ell_2$-norm. We conclude by noting that model behavior can also be studied through other suitable accumulations of the influence tensor $\Gamma$, which we leave to future work.

### 4.1. Validation through parameter pruning

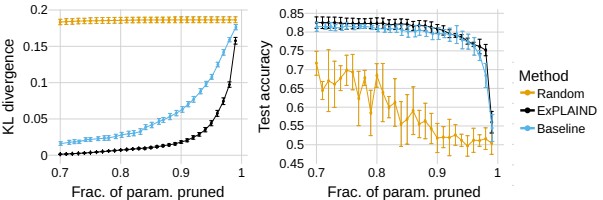

*Figure 2.* We prune the CNN weights and achieve test accuracy comparable to Li et al. (2017). Across all sparsity levels, the KL divergence of our model outputs is consistently lower. We report the means and standard deviations over 5 runs.

To validate our parameter importance scores, we conduct pruning experiments on the CNN model, ranking each parameter $\theta$ by its kernel importance score $\Psi_S(\theta)$. For a fraction $c$ and number of parameters $D$, we only keep the TOP-$cD$ parameters and set the rest to zero. To retain the model's ability to predict, we do not prune the output layer. To contextualize our results, we compare against an implementation of Li et al. (2017), a popular pruning method proposed for CNN models, which is based on an iterative procedure in which the model is retrained in each step after pruning weights by magnitude importance.

As shown in Figure 2, we find that our approach performs competitively against Li et al. (2017)'s approach on sparsity levels from 70% to 99%. Furthermore, our score-based, training-free pruning replicates the original model more closely than the baseline as is evidenced by the much lower KL divergences over all levels of sparsity. This underlines that our influence scores are indeed able to accurately quantify the influence of the parameters on the model's predictions. Note that this comparison is to validate that our method finds meaningful influence scores and does not aim to frame ExPLAIND as a SOTA pruning method.

### 4.2. Efficiency

Recall that all explanations provided rely on accumulating slices of the influence tensors $\Gamma$, which affects the efficiency of the resulting computations. In general, materializing all scores is expensive, as the memory complexity alone is in $\mathcal{O}(NDMO)$ for $N$ training steps, $D$ parameters, $M$ training samples, and $O$ output dimensions. Through its flexibility to support arbitrary levels of granularity on these dimensions, ExPLAIND gives an interface to control the trade-off between high and low granularity of explanations, i.e. whether they are local or global along the aforementioned dimensions. This also affects complexity: For exam-

ple, if one is interested only in the influence of components over the training trajectory, one can *accumulate the gradients* in the train feature map over the training data before computing the dot products (proof in Appendix F.4), thus reducing complexity.

Further, large model sizes make it infeasible to store parameter checkpoints at each step. Subsampling the update steps is a second possibility to improve efficiency. We test the the sensitivity of this technique on the MNIST model: We rank the update steps by the absolute loss change on a batch of held-out samples. We then compute the predictions of a decomposition based on the TOP-$X$ steps for different levels of sparsity $X$ (details in Appendix G.2). We find that the reconstructed prediction of a subsample of steps is close to the actual model prediction in terms of accuracy from around a sparsity of 60% (full plot in Appendix Figure 8). In other words, for our MNIST model the number of steps considered can almost be halved and the resulting decomposition is still accurate.

In Section 6, we apply these two strategies and show that they enable efficient and effective scaling to LLMs.

## 5. Case study I: Grokking 'ExPLAIND'

Grokking is a training phenomenon where models first memorize, and then, after prolonged training, suddenly generalize (Power et al., 2022). Nanda et al. (2023) argue for multiple learning stages: (1) *memorization of the training data*, (2) *circuit formation*, in which the model learns a robust mechanism and thus generalizes, and (3) *cleanup*, during which the regularization "removes" memorization components. Other evidence emphasizes the role of dataset size, model capacity, and regularization (Huang et al., 2024; Wang et al., 2024). We use ExPLAIND to revisit these hypotheses, integrating perspectives on model components, training data, and training dynamics in a single analysis. We further verify our insights through ablation experiments.

Our testbed is the well-studied modulo addition task by Varma et al. (2023) where the samples are of the form $[a][+][b][\mathrm{mod}113 =]$. We train a Transformer with AdamW (details in Appendix E.2) to predict the correct result. As shown in the top-left of Figure 3a, the model exhibits Grokking: training accuracy rises sharply, reaching nearly perfect accuracy around step 800. During this phase, test accuracy remains close to zero, but subsequently increases until 100% at step 1939, at which point training is stopped.

### 5.1. Influence decomposition

To study Grokking with ExPLAIND, we begin by decomposing the model's predictions into kernel and regularization influences over the training trajectory (see Figure 3a). We then study their layer-wise decompositions. This provides a

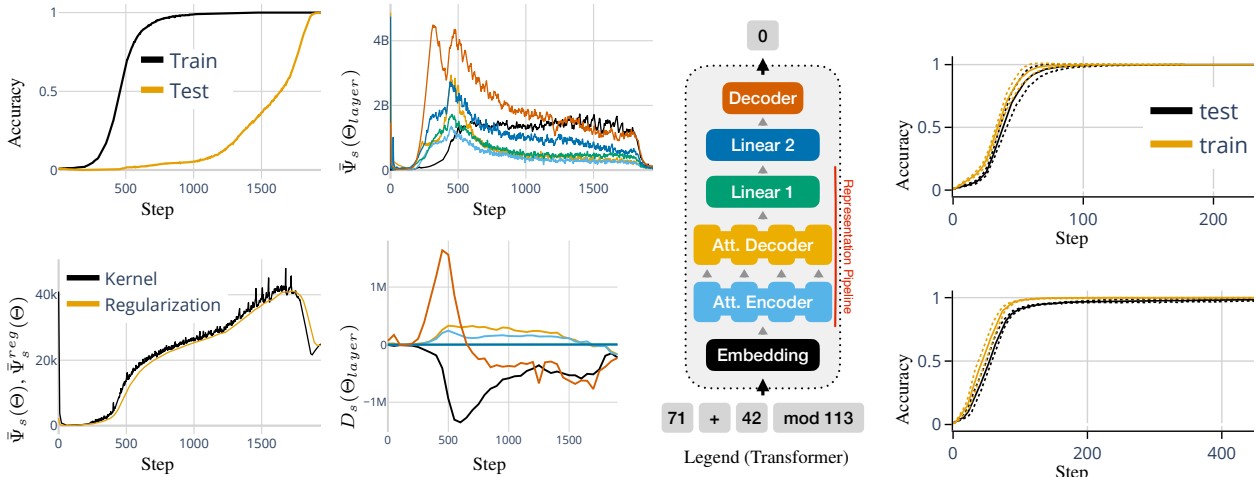

*(a)* **Left:** Accuracy and importances of kernel and regularization over the training of the Transformer model. **Right:** Layer-wise importance of the kernel (top) and influence of the regularization (bottom). As shown in the legend on the right, the Transformer consists of an attention layer (Att. Encoder and Att. Decoder) and an MLP (Linear 1 and Linear 2).

*(b)* We train a model initialized with the representation pipeline from the final model (**top**) and from step 1539 (**bottom**) and initialize the rest at random.

*Figure 3.* Training statistics, influences, and validation of our representation pipeline hypothesis.

global view of the training dynamics before, during, and after generalization, and highlights which model components drive these phases. ExPLAIND reveals the following trends in how kernel and regularization influences evolve:

**Kernel versus regularization balance.** The absolute influences $\bar{\Psi}_s(\Theta)$ and $\bar{\Psi}_s^{reg}(\Theta)$ (bottom left of Figure 3a) grow together from the start of memorization (around epoch 200), both peaking near step 1700 before declining. In the final phase, $\bar{\Psi}_s^{reg}(\Theta)$ dominates which indicates that the final training phase is governed by a higher relative influence of regularization. On the layer level, the memorization phase coincides with a peak in regularization influence $D_s(\Theta_{layer})$ (bottom right), and absolute influences $\bar{\Psi}_s(\Theta_{layer})$ (top right) of the decoder. This establishes the decoder as the most influential component for memorization.

**Middle-layer alternation and circuit formation.** The second peak of the decoder influence (top right) is preceded by peaks of the attention and linear layers, suggesting a single alternation between fitting the decoder and the intermediate layers. This marks the beginning of the circuit formation and hints at a representation pipeline forming in the middle layers.[3] Their influence then decreases, supporting this interpretation. With respect to regularization, both layers have a rather low influence, indicating that regularization acts primarily on the embedding and decoder layers.

**Late embedding dominance.** The embedding layer only begins to influence predictions in the circuit formation phase,

evidenced by its sharp relative regularization dominance in $D_s(\Theta_{layer})$ and its rise in $\bar{\Psi}_s(\Theta_{layer})$. Remarkably, the latter continues to increase slightly during the rest of the training while all other layers' influences decrease.

These insights suggest that the model progresses through three influence phases: decoder-driven memorization, middle-layer alternation for circuit formation, and eventual embedding-decoder dominance under higher regularization influence. The rapid increase in test accuracy can thus be explained by a *simultaneous alignment of the input embeddings and decoder around the representation pipeline in the intermediate layers* (marked red in Figure 3a).

To confirm that the alignment around this representation pipeline is causal, we train models initialized at random but replace only the Attention and Linear-1 layers with their grokked counterparts, which we hypothesize to be central to the representation pipeline. As shown in Figure 3b, this initialization leads to instant generalization within the first 200 training steps, and, in particular, bypasses the usual memorization phase. Additional experiments (Appendix Figure 11) confirm that even using only the attention layer produces the same phenomenon. Starting from earlier checkpoints also leads to rapid generalization, albeit with slightly lower final accuracy. This suggests a continuous refinement of the representation pipeline during the circuit formation phase. To further probe this effect, we swap grokked layers into intermediate checkpoints (see Appendix Figure 11). The outer layers identified above consistently improve the predictions of these checkpoints, whereas other combinations decrease performance. This shows that the aligned final embedding and decoder layers can already operate ef-

---

[3]This interpretation is further supported by the simultaneous emergence of cyclic patterns on the influence level around steps 450 - 500 (see Section 5.2 and Appendix Figure 13).

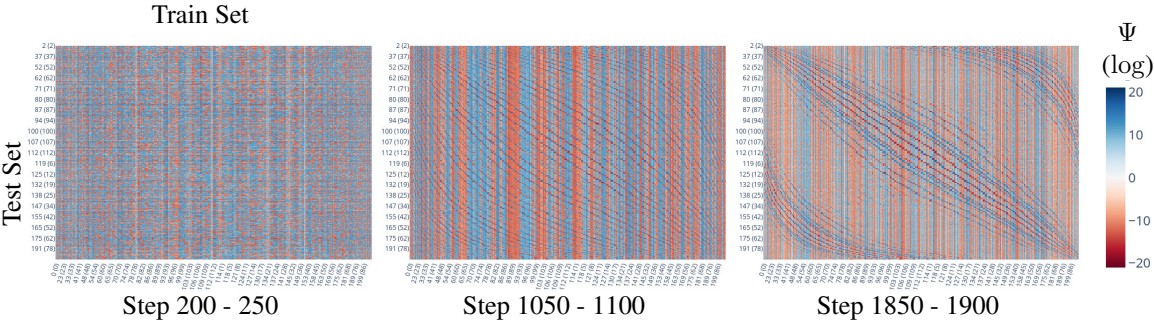

*Figure 4.* At different training stages, we show the influences $\Psi_S(\Theta_{dec}, x, x')$ of the training examples $x'$ on the test samples $x$ of the decoder layer $\Theta_{dec}$, summed over the output dimensions, for predictions on the test set and the training set, accumulated over the preceding 50 steps. The plots are labeled and ordered by the sum of inputs $a + b$ and the corresponding result ($a + b \bmod 113$). Corresponding figures for the other layers are provided in the Appendix in Figure 9.

fectively around a pipeline that is not yet fully generalized.

These findings refine the three-stage description of Grokking by Nanda et al. (2023) by showing that in the final phase the model outer' layers align around a generalized representation pipeline. They also connect to the efficiency perspective by Huang et al. (2024): our findings suggest that once robust representations have formed, the influence of the regularization suppresses inefficient memorization solutions.

In the next section we show that this representation pipeline encodes a cyclic data geometry that becomes increasingly robust during training.

### 5.2. Cyclic geometry

Next, we apply ExPLAIND from the data perspective to examine what structure the model learns. We study two complementary objects: $\Psi_s(\Theta_{dec}, x, x')$ (Figure 4) capturing how training samples $x'$ influence predictions $x$ with respect to the decoder and the *similarity matrices* $Sim_{\Theta_{layer}}(x, x')$ (Figure 5) which capture how samples are represented relative to each other. We find:

**Emergence of cyclic patterns in the kernel.** The vertical bands in Figure 4 reveal that at all training stages, each training sample has a global influence. After memorization, off-diagonal patterns emerge and sharpen, aligning with modular equivalence classes ($a + b \bmod 113$). Initially, these cycles have high frequency (about 2), i.e. the influence is strong for training samples whose label differences are a multiple of 2. Later in training, this cyclic geometry continuously shifts towards lower frequencies.

**Generalizable data geometry.** Similarity matrices (Figure 5) suggest a similar emergence of cyclic patterns, especially in the higher layers of the model. This indicates that the model indeed learns to represent the samples in a space where similarity is approximately a cosine of frequency 113. We test this hypothesis by fitting a Lasso regression to the

similarity matrix accumulated over epochs 1850 to 1900 of the decoder, where we take as input features all cosines and sines of the pairwise differences of frequencies 2 to 113. The result is shown in Appendix Figure 12 and confirms our qualitative observation with the coefficient of the cosine of frequency 113 being by far the largest. Furthermore, we note that in the final model this representation emerges only in the attention decoders, suggesting that the model first needs to combine the two input numbers and then proceeds to refining the cyclic geometry observed in higher layers.

Revisiting our layer-swapping experiment, we compare the confusion matrices of the swapped and original checkpoints (see Appendix Figure 10). We observe that the systematic cyclic off-diagonal error patterns are reduced once the final model's outer layers are swapped in, suggesting that the embedding-decoder alignment indeed changes the prediction algorithm to one that uses the cyclic representations.

Our observations agree with Nanda et al. (2023), who find that neurons in the linear layers are computing combinations different cosines and sines. As ExPLAIND reveals, neuron-level mechanisms that they describe result in a cyclic global pattern that is simple to interpret and the result of a continuous refinement from cycles of higher to lower frequency. In sum, our results show that Grokking in the modulo Transformer reflects the progressive refinement of a cyclic data geometry and its alignment with input and output layers.

## 6. Case study II: LLM Pretraining

So far, we chose to focus on validating and studying ExPLAIND in smaller, interesting scenarios. In this section, we discuss ExPLAIND's applicability to larger scenarios and show initial evidence that its decomposition into influence scores is accurate and helps attributing LLM behavior. For this, we choose to study `EuroLLM-1.7B` (Martins et al., 2024), a 1.7B parameter model based on the Llama architecture (et al., 2024) trained on 4T tokens.

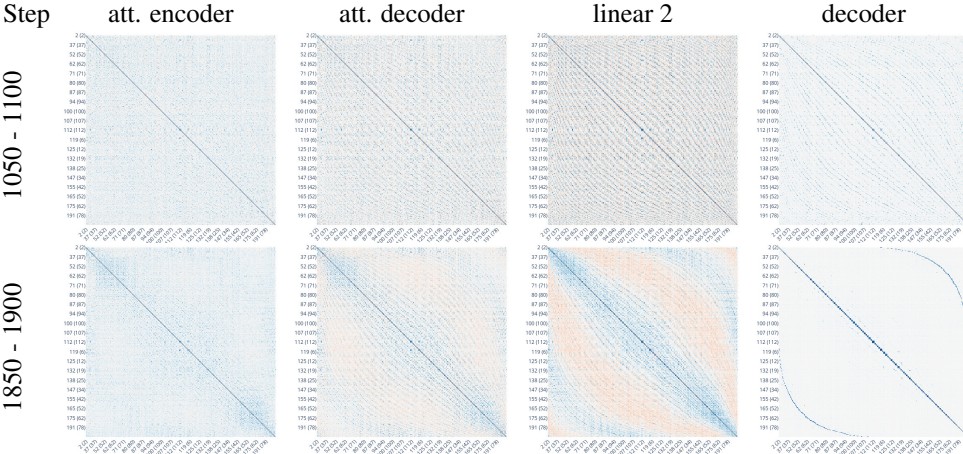

*Figure 5.* Similarity $Sim_{\Theta_{layer}}(x, x')$ of predictions of the test set of the Transformer model accumulated over different training stages. All layers and other training stages shown in Appendix Figure 13.

**Scaling to larger settings.** As outlined in Section 4.2, we rely on (1) *(early) accumulation* of the training feature map, and (2) *subsampling the number of training steps*. We use these principles to decompose the loss trajectory of EuroLLM. Note that we do not need to retrain the model: We can view the 37 checkpoints over the first training phase that are available to us as a subsampling of the training steps, i.e. an application of principle (2). Applying principle (1), we do not decompose single training samples (i.e. single next-token predictions), but sequences of length $4096$, which enables us to efficiently sum up the gradients before the dot product of the feature maps without loss of accuracy. We also don't need access to the full training data: Since we integrate between the parameters of subsequent steps, we sample a single batch with the same composition as the original batches (3072 sequences sampled from 35 languages) and perform one update step for each checkpoint.

For computing the scores, we again use $100$ integration steps in the test feature map. We are able to decompose the loss change of an unseen batch drawn from the same distribution as the simulated training batch at each checkpoint in about 15 minutes on a single Nvidia H100 GPU. Despite the much larger scale than our previous experiments, the simulated loss trajectory is replicated accurately by the resulting influence scores. Over the checkpoints, the mean loss change due to our training batch is at $-6.05 \cdot 10^{-4}$ which is reproduced up to an average error of $4.46 \cdot 10^{-8}$ with a standard deviation of $3.12 \cdot 10^{-8}$ over the checkpoints. This result shows that ExPLAIND can be used to study larger scenarios than the ones we chose to study above. We will now use the resulting scores to study EuroLLM's pretraining dynamics.

**Training dynamics.** We use the computed decomposition to study the global, parameter-level training dynamics of EuroLLM. We accumulate over the data-dimension like before and plot the resulting influence scores grouped by two different layer views in Figure 6. Recall that in this case study we decompose the loss trajectory of the model. Negative values on this scale indicate loss decrease, i.e. learning success on the test examples we decompose over. We find a surprising continuity in the parameter-wise influences over the training, which we observe to follow two remarkably distinct phases. First, the model's optimizations mainly stem from changes in the MLP layers (see left plot), first mainly in layers close to the input and then in layers closer to the output. This trend reverses around step 60K after which the relative influence of intermediate and lower layers becomes stronger. Further, loss-decreasing influences are attributed to the attention layers in this second phase. Interestingly, the model embeddings only play a small role in these views.

We also plot the influence of the regularization in Appendix Figure 14. We observe two trend changes at a similar mark: The magnitude of the influences increases sharply until around 50K after which their change decelerates. In addition, the regularization influence through the output embeddings changes from being negative to positive after step 60K, indicating that effects of regularization on them first decreases then increases the loss in the first and second phase respectively. We hypothesize that this two-phase trajectory reflects a shift from early representation building dominated by MLP updates to later context integration and distributed refinement involving stronger attention contributions.

In contrast, the influences of the first moment which are separated from the batch influences in this setting (details in Appendix F.4) show an oscillatory changing their sign between each step (see Appendix Figure 15). This corroborates previously reported, non-smooth learning dynamics of deep networks, by which the loss tends to be non-monotonous locally but to decrease over wider windows (Cohen et al., 2021) and suggests that the first moment is one source of such oscillations in LLMs trained via AdamW.

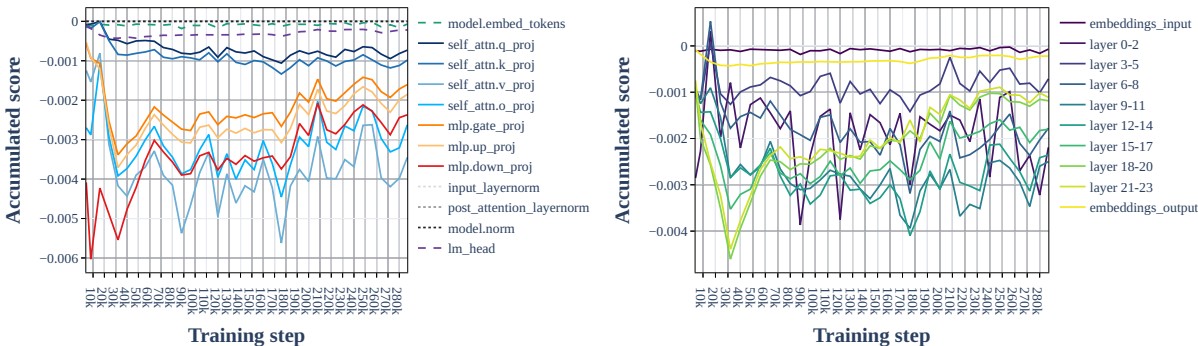

*Figure 6.* Accumulated influence scores $\Phi(s, \Theta)$ based on the decomposition of the loss trajectory of EuroLLM pretraining by parameter type (left) and layer depth (right). We find that EuroLLM follows two learning phases: At first, especially MLPs in the layers close to the outputs seem to drive learning. Step $60k$ marks a phase transition: The influence of intermediate and layers closer to the inputs increases, while the upper layers' relative influence decreases. Further, relative learning influences shift from the MLP layers to the attention layers.

## 7. Discussion and Conclusion

ExPLAIND offers a unified framework that bridges model components, data, and training dynamics, addressing a gap in post-hoc interpretability. Building on gradient path theory, it extends the Exact Path Kernel to realistic optimization regimes which is of independent interest. We validate the EPK representation and demonstrate the effectiveness of the resulting scores in parameter pruning. Through its theoretical foundation, ExPLAIND provides additive parameter-wise influence scores that can be aggregated to different levels of granularity and viewed from multiple perspectives. This positions ExPLAIND as a useful toolbox for unified attribution of model behavior.

Our exploratory study on Grokking highlights the utility of ExPLAIND by uncovering a novel perspective on its learning phases, with a central role for alignment and capability reuse. In particular, we identify an alignment phase characterized by the high relative influence of regularization on the outer layers preceded by the building of a representation pipeline. This suggests that prolonged training serves to refine and re-use existing representations rather than build new ones. Future work should be dedicated to the question of whether our insights in the modulo Transformer generalize to larger models and more complex tasks.

In our second case study of the global training dynamics of EuroLLM we find an interesting two-phase trajectory, which first identifies upper MLP learning to be responsible for large fractions of the loss decrease and then intermediate attention layers. Further, regularization influence in the output embeddings shifts from loss decreasing to loss increasing during this transition. This study paves the way for future, more detailed inquiries into larger settings, especially understanding the latent mechanisms of LLMs.

More broadly, our results indicate that attributions to data and model components vary substantially across training,

with critical patterns emerging at specific stages. Future interpretability methodology should therefore be designed to better surface these critical stages.

As regards potential limitations, the ExPLAIND framework focuses on parameter-level analysis and is not designed to provide causal explanations. The pruning results indicate that a level of counterfactual behavior can be predicted from our scores. Future work should investigate whether ExPLAIND can identify activation-level objects like mechanistic circuits (Olah et al., 2020).

Previous work on data attribution is also framed in terms of predicting the behavior of counterfactual models trained on subsets of the data (Koh & Liang, 2017). The presented study of the Transformer model and Bell et al. (2023)'s work provide initial evidence for ExPLAIND's applicability on the level of attributing training data, but not in such a causal way. As a first inquiry, we investigate whether our scores also perform in terms of causal data attribution on the presented CIFAR and MNIST settings. As we detail in Appendix G.1, the results are not as promising as for parameter attribution: While all baselines we evaluate do not perform well in these settings, data influence scores naively derived from ExPLAIND perform roughly on par with TracIn and are slightly outperformed by TRAK. Future work should investigate other ways to compare against other methods—both unified and non-unified.

The cost of applying ExPLAIND in practice is a further limitation, which we adress in Section 4.2. Nevertheless, even when optimizing its application in the ways we discuss there, the overhead can be substantial. Other directions for the efficiency of ExPLAIND should be investigated, e.g., learned approximations of accumulated scores.

In conclusion, this work establishes a promising novel avenue for studying learning dynamics, training data, and model internals of neural networks in a unified framework.

## Acknowledgements

The members of MaiNLP provided valuable inspiration and feedback for this project. In particular, we wish to thank Felicia Körner, Xinpeng Wang, Verena Blaschke, Silvia Casola, Shijia Zhou, and Domenico De Cristofaro for their feedback on previous versions of this work. We also extend special thanks to Marwan Elsayed for his detailed feedback and corrections. Furthermore, we acknowledge the support provided to BP through the ERC Consolidator Grant DIALECT 101043235. MM is supported by the DAAD programme Konrad Zuse Schools of Excellence in Artificial Intelligence, sponsored by the German Federal Ministry of Research, Technology and Space.

## Impact Statement

This paper presents work whose goal is to advance the field of Machine Learning. There are many potential societal consequences of our work, none which we feel must be specifically highlighted here. However, we include an extensive discussion of additional ethical considerations in Appendix C.

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

# Supplement to the paper "ExPLAIND: Unifying Model, Data, and Training Attribution to Study Model Behavior"

## A. Reproducibility Statement

The experimental methodology is described in detail in Appendix E, and all experiments are fully reproducible. Source code will be released upon acceptance and is also provided as part of the supplementary material. The proof of the main statement, Theorem 3.1, and Corollary 3.2 is included in Appendix F, the proofs of the remaining statements are part of the main text.

## B. LLM Usage Statement

We used large language models (LLMs) for editing the manuscript, including for grammar, spelling, and rephrasing. We further use LLMs for support with coding. For both, we made sure to check the validity and security of all LLM outputs. AI tools do not contribute substantively to the ideas, research contributions, or results.

## C. Ethical Considerations and Broader Impacts

**Interpretability for Fairness and Accountability.**   Interpretability is a foundational requirement for building machine learning systems that are transparent, trustworthy, and legally accountable. Our framework, ExPLAIND, contributes to this goal by offering explanations that connect model behavior back to training data and model components. This is especially important in high-stakes domains (e.g. healthcare, criminal justice, finance), where decisions made by machine learning models must be auditable and understandable. Transparent systems are essential for identifying and mitigating biases, ensuring compliance with regulatory standards, and enabling meaningful human oversight.

**Causality, Overinterpretation, and Misleading Explanations.**   Although ExPLAIND provides rigorous weight-level influence scores, they are inherently statistical and not causal. Misinterpreting these scores as direct causal claims about model behavior could lead to incorrect conclusions or misguided policy decisions. Practitioners and researchers must exercise caution when drawing inferences from post-hoc explanations and should clearly communicate the implications and limitations of an explanation.

**Respecting Data Ownership.**   Recent investigations have revealed that major AI companies have utilized large-scale datasets containing pirated content, such as Library Genesis (LibGen), to train their models without obtaining permission from the original authors or rights holders. This practice not only infringes upon the intellectual property rights of creators but also raises significant ethical concerns regarding consent and fair compensation. Theoretically grounded attribution of training data and model components like ExPLAIND opens the door for mechanisms that acknowledge, attribute, and compensate the creators of influential data, thus respecting intellectual property rights.

**Unequal Access to Computational Resources.**   The development and application of computationally intensive interpretability methods, such as ExPLAIND, underscore a significant ethical concern: the disparity in access to necessary computational resources. This "compute divide" disproportionately favors well-funded industry players and elite academic institutions, enabling them to conduct advanced AI research and model auditing. In contrast, smaller institutions and independent researchers often lack the resources to engage in such work, limiting their participation in critical areas such as model interpretability and accountability. This imbalance not only hampers diverse contributions to the field but also raises concerns about whose models are scrutinized and whose voices are heard in shaping AI's future.

**Environmental Costs and the Role of Efficient Interpretability.**   Training large models, and by extension applying post-hoc interpretability methods like ExPLAIND, comes with significant computational and environmental costs. While our method is computationally expensive — often comparable to a single training run — we argue that this cost is justified in contexts where theoretical robustness and faithful attribution are necessary. Nonetheless, we acknowledge the environmental impact and advocate for minimizing computational overhead through algorithmic optimization, more efficient implementations and minimizing redundant applications. Future work should investigate scalable approximations of ExPLAIND to reduce emissions while preserving interpretability guarantees.

# D. Extended Literature Review

This section provides an extended version of the literature review (see 2), including additional material relevant to the present work that was omitted from the main paper due to space constraints.

**Post-hoc interpretability.** There are many more approaches to post-hoc interpretability methodology that fall into one the three traditional explainability types, *input feature attribution* (Ribeiro et al., 2016; Lundberg & Lee, 2017; Binder et al., 2016; Zeiler & Fergus, 2014), the *training data attribution* (Park et al., 2023; Grosse et al., 2023; Chen et al., 2021; Ilyas et al., 2022; Bae et al., 2024; Liu et al., 2025; Ghorbani & Zou, 2019; Koh & Liang, 2017), and *model component attribution* (Tenney et al., 2019; Wiegreffe & Pinter, 2019; Vig et al., 2020; Nanda, 2023; Arditi et al., 2024; Tang et al., 2024; Olah et al., 2020; Elhage et al., 2022; Rai et al., 2024).

**Grokking.** Grokking refers to a training phenomenon where models initially overfit but eventually generalize after prolonged training (Power et al., 2022). Liu et al. (2023) expanded this study to a broader suite of tasks and model architectures, providing a systematic characterization of Grokking's occurrence. More recent work (Wang et al., 2024; Zhu et al., 2024; Huang et al., 2024) explores the implicit reasoning capabilities that arise during Grokking, the critical role of dataset size, and their connection to the double descent phenomenon. Nanda et al. (2023) argue that Grokking occurs in three phases: memorization, circuit formation, and cleanup. Our work refines this narrative, instead suggesting a progression through memorization, representation pipeline formation, and embedding-decoder alignment.

# E. Technical Details and Hyperparameters

In the next two sections, we specify the technical details of our models and data, as well as the hyperparameters we use. All implementation is provided in the supplementary material. In Section E.3 we detail the computation resources we used.

## E.1. CNN Models

**CIFAR-2 model.**   We train a ResNet 9 model (He et al., 2016) with with 5 layers and two residual blocks, each consisting of two additional convolution layers with max-pooling, ReLU activations and a logarithmic softmax over the two dimensional output. We take the CIFAR-2 subset of CIFAR-10 (Krizhevsky et al., 2014) consisting of the classes dog and cat (10000 samples) and train using SGD with momentum of 0.9 for 12 epochs with a mini-batch size of 256 and weight decay of 0.005. We use a learning rate schedule that peaks in epoch 5 at 0.1. The loss is a cross entropy loss assuming logarithmic probability inputs.

**MNIST model.**   We train a four-layer CNN model on a subset of 5120 samples of MNIST (**?**), a standard setting for handwritten digit recognition. The model consists of two convolution blocks, each comprised of a convolution layer, max pooling, and a ReLU activation, and two fully connected layers where the first is followed by another ReLU activation and the final output is gained via a softmax over the ten output dimensions, which serves as the input of the cross-entropy loss target. The model is trained using AdamW with betas $\beta = (0.7, 0.9)$, weight decay of $\lambda = 0.1$, and a constant learning rate of $\gamma = 0.01$ (the remaining hyperparameters are chosen as the `torch` defaults). Further, we use a batch size of 256 and train for a single epoch. The resulting model achieves $93.55\%$ accuracy on a test set of 1024 samples.

## E.2. Transformer Model

The Transformer model, which was proposed by Varma et al. (2023) and used by Nanda et al. (2023), has a single layer encoder as described by Vaswani et al. (2017) and a *decoder* that consists of a single, fully connected layer mapping from the hidden dimension of 64 to the 115-token vocabulary. We use a 115-token input *embedding* without positional encoding, followed by a multi-head attention layer with four heads, each mapping to a space of dimension 16. We refer to the modules mapping to the lower dimensional spaces, that are used to compute the attention scores, as *attention encoder*, and accordingly call the modules reading from the representations after applying attention as *attention decoder*. The MLP layer on top of that consists of two fully connected layers (*Linear 1* and *Linear 2*), which map to and read from a 512-dimensional latent space. We visualize the transformer in the legend of Figure 3a.

We train on full batches using AdamW with a fixed learning rate of 0.001, weight decay of 4.0, and $\beta_1 = 0.98$, $\beta_2 = 0.99$ for the scaling parameters of the first and second moment estimates of the gradient, respectively.

The dataset consists of 4000 samples which each contain four tokens, namely the number [a], an addition token [+],

the number [b] and the token [mod 113 =]. Here, $a, b \in \{0, 1, ..., 112\}$ and we always enforce $a \geq b$, leading to a total number of $\frac{113 \cdot 112}{2} = 6328$ possible data points of which we include $4000$ randomly sampled ones in the train set and another $2000$ in the test set labeled with the correct output token [c] containing the correct result $c = (a + b) \mod 113$ which has to be predicted.

### E.3. Compute resources used in our experiments

Model training and retraining were carried out on a 20GB partition of NVIDIA A100 GPU for a total of less than 5 hours. Applying ExPLAIND to both models was much more compute intensive, resulting in about 20 hours of computation on a H200 GPU with 140GB GPU-RAM. Debugging and running the ablations presented, we carried out 12 such full runs of the EPK predictions computing ExPLAIND influence scores, leading to a total of about 240 H200 GPU-hours.

## F. Proofs and mathematical details

### F.1. Proof of the decomposition for AdamW

We first restate the extended Theorem with the full set of assumptions.

**Statement** (Kernel Equivalence for AdamW, extended version of Theorem 3.1 in main text). *Let $f_\theta : \mathcal{X} \to \mathcal{Y}$ be a model with parameters $\theta \in \mathbb{R}^D$, where $\mathcal{X} \subseteq \mathbb{R}^I$ and $\mathcal{Y} \subseteq \mathbb{R}^O$. Let $L : \mathcal{Y} \times \mathcal{Y} \to \mathbb{R}_{\geq 0}$ be a per-sample loss. Assume that, for every $x \in \mathcal{X}$, the map $\theta \mapsto f_\theta(x)$ is continuously differentiable, and that, for every $y \in \mathcal{Y}$, the map $z \mapsto L(z, y)$ is continuously differentiable. Let $\mathcal{D} = \{(x_1, y_1), \ldots, (x_M, y_M)\}$ be a dataset. For each training step $i \in \{0, \ldots, N-1\}$, let $B_i \subseteq 1, \ldots, M$ denote the mini-batch used at step $i$, and define the batch loss*

$$\mathcal{L}_i(\theta) := \sum_{k=1}^{M} \mathbf{1}_{k \in B_i} w_{i,k} L(f_\theta(x_k), y_k),$$

*where $w_{i,k} \in \mathbb{R}$ denotes the weight assigned to sample $(x_k, y_k)$ at step $i$. Suppose that $\theta_N$ is obtained from $\theta_0$ by $N$ steps of AdamW with learning rates $\alpha_s \in \mathbb{R}_{>0}$, weight decay $\lambda \in \mathbb{R}_{\geq 0}$, and updates*

$$\theta_{s+1} = \theta_s - \alpha_s \frac{m_{s+1}}{(1 - \beta_1^{s+1})(\sqrt{\hat{v}_{s+1}} + \epsilon)} - \alpha_s \lambda \theta_s, \qquad s = 0, \ldots, N-1,$$

*where*

$$m_{s+1} = \beta_1 m_s + (1 - \beta_1)\nabla_\theta \mathcal{L}_s(\theta_s), \qquad v_{s+1} = \beta_2 v_s + (1 - \beta_2)\big(\nabla_\theta \mathcal{L}_s(\theta_s)\big)^2, \qquad \hat{v}_{s+1} := \frac{v_{s+1}}{1 - \beta_2^{s+1}}$$

*with $m_0 = v_0 = 0$, $\epsilon \in \mathbb{R}_{>0}$, and where powers, square roots, and division involving vectors are understood coordinatewise. Then, for every $x \in \mathcal{X}$,*

$$f_{\theta_N}(x) = f_{\theta_0}(x) - \sum_{k=1}^{M} \sum_{s=0}^{N-1} \phi_s^{test}(x) \, \phi_s^{train}(x_k) - \sum_{s=0}^{N-1} \phi_s^{test}(x) \, \mathbf{r}_s, \tag{3}$$

*where*

$$\phi_s^{test}(x) := \int_0^1 \nabla_\theta f_{\theta_s(t)}(x) \, dt \in \mathbb{R}^{O \times D},$$

$$\phi_s^{train}(x_k) := \sum_{i=0}^{s} \alpha_{s,i} \, \mathbf{1}_{k \in B_i} w_{i,k} \frac{\nabla_\theta \big(L(f_{\theta_i}(x_k), y_k)\big)}{\sqrt{\hat{v}_{s+1}} + \epsilon} \in \mathbb{R}^D,$$

$$\mathbf{r}_s := \alpha_s \lambda \theta_s \in \mathbb{R}^D,$$

$$\theta_s(t) := \theta_s + t(\theta_{s+1} - \theta_s) \in \mathbb{R}^D,$$

$$\alpha_{s,i} := \alpha_s (1 - \beta_1) \beta_1^{s-i} (1 - \beta_1^{s+1})^{-1} \in \mathbb{R}.$$

*Proof.* Let $x \in \mathcal{X}$. For each $s = 0, \ldots, N-1$, define the interpolation of parameters at step $s$ and $s + 1$ as

$$\theta_s(t) := \theta_s + t(\theta_{s+1} - \theta_s), \qquad t \in [0, 1].$$

Then

$$\frac{d\theta_s(t)}{dt} = \theta_{s+1} - \theta_s.$$

By the AdamW update rule,

$$\theta_{s+1} = \theta_s - \alpha_s \frac{m_{s+1}}{(1 - \beta_1^{s+1})(\sqrt{\hat{v}_{s+1}} + \epsilon)} - \alpha_s \lambda \theta_s,$$

where $\epsilon \in \mathbb{R}_{>0}$, $\hat{v}_{s+1} = v_{s+1}/(1 - \beta_2^{s+1})$, and powers, square roots, and division involving vectors are understood coordinatewise. Using $m_0 = v_0 = 0$ and unrolling the momentum recursion gives

$$m_{s+1} = \beta_1 \cdot m_s + (1 - \beta_1) \cdot \nabla_\theta \mathcal{L}_s(\theta_s) = \sum_{i=0}^{s} (1 - \beta_1)\beta_1^{s-i} \nabla_\theta \mathcal{L}_i(\theta_i).$$

Moreover,

$$v_{s+1} = \beta_2 v_s + (1 - \beta_2)\big(\nabla_\theta \mathcal{L}_s(\theta_s)\big)^2,$$

where the square is understood coordinatewise. We do not unroll this recursion. Therefore,

$$\frac{d\theta_s(t)}{dt} = -\alpha_s \frac{m_{s+1}}{(1 - \beta_1^{s+1})(\sqrt{\hat{v}_{s+1}} + \epsilon)} - \alpha_s \lambda \theta_s.$$

Substituting the expression for $m_{s+1}$ yields

$$\frac{d\theta_s(t)}{dt} = -\sum_{i=0}^{s} \alpha_{s,i} \frac{\nabla_\theta \mathcal{L}_i(\theta_i)}{\sqrt{\hat{v}_{s+1}} + \epsilon} - \alpha_s \lambda \theta_s,$$

where

$$\alpha_{s,i} := \alpha_s (1 - \beta_1)\beta_1^{s-i}(1 - \beta_1^{s+1})^{-1}.$$

By definition of the batch loss,

$$\mathcal{L}_i(\theta) = \sum_{k=1}^{M} \mathbf{1}_{k \in B_i} \, w_{i,k} \, L(f_\theta(x_k), y_k),$$

and therefore

$$\nabla_\theta \mathcal{L}_i(\theta_i) = \sum_{k=1}^{M} \mathbf{1}_{k \in B_i} \, w_{i,k} \, \nabla_\theta\big(L(f_{\theta_i}(x_k), y_k)\big).$$

Hence

$$\frac{d\theta_s(t)}{dt} = -\sum_{i=0}^{s} \sum_{k=1}^{M} \alpha_{s,i} \, \mathbf{1}_{k \in B_i} \, w_{i,k} \, \frac{\nabla_\theta\big(L(f_{\theta_i}(x_k), y_k)\big)}{\sqrt{\hat{v}_{s+1}} + \epsilon} - \alpha_s \lambda \theta_s.$$

Since $\theta \mapsto f_\theta(x)$ is continuously differentiable and $\theta_s(t)$ is affine in $t$, the chain rule applies along the path $t \mapsto \theta_s(t)$. Applying it, we thus obtain

$$\frac{d}{dt} f_{\theta_s(t)}(x) = \nabla_\theta f_{\theta_s(t)}(x) \frac{d\theta_s(t)}{dt}.$$

Substituting the expression for $\frac{d\theta_s(t)}{dt}$ gives

$$\frac{d}{dt} f_{\theta_s(t)}(x) = -\sum_{i=0}^{s} \sum_{k=1}^{M} \alpha_{s,i} \, \mathbf{1}_{k \in B_i} \, w_{i,k} \, \nabla_\theta f_{\theta_s(t)}(x) \frac{\nabla_\theta\big(L(f_{\theta_i}(x_k), y_k)\big)}{\sqrt{\hat{v}_{s+1}} + \epsilon} - \alpha_s \lambda \nabla_\theta f_{\theta_s(t)}(x)\theta_s.$$

Since the sums over $i$ and $k$ are finite, they may be interchanged with the integral. Integrating from $t = 0$ to $t = 1$ and using the fundamental theorem of calculus yields

$$\begin{aligned}
f_{\theta_{s+1}}(x) - f_{\theta_s}(x) = &-\sum_{k=1}^{M} \left(\int_0^1 \nabla_\theta f_{\theta_s(t)}(x)\, dt\right) \left(\sum_{i=0}^{s} \alpha_{s,i} \, \mathbf{1}_{k \in B_i} \, w_{i,k} \, \frac{\nabla_\theta\big(L(f_{\theta_i}(x_k), y_k)\big)}{\sqrt{\hat{v}_{s+1}} + \epsilon}\right) \\
&- \alpha_s \lambda \left(\int_0^1 \nabla_\theta f_{\theta_s(t)}(x)\, dt\right)\theta_s.
\end{aligned} \tag{4}$$

Summing over $s = 0, \ldots, N-1$, we obtain

$$f_{\theta_N}(x) = f_{\theta_0}(x) + \sum_{s=0}^{N-1} \left( f_{\theta_{s+1}}(x) - f_{\theta_s}(x) \right).$$

Substituting the previous identity yields

$$
f_{\theta_N}(x) = f_{\theta_0}(x) - \sum_{s=0}^{N-1} \sum_{k=1}^{M} \left( \int_0^1 \nabla_\theta f_{\theta_s(t)}(x) \, dt \right) \left( \sum_{i=0}^{s} \alpha_{s,i} \, \mathbf{1}_{k \in B_i} \, w_{i,k} \, \frac{\nabla_\theta \big( L(f_{\theta_i}(x_k), y_k) \big)}{\sqrt{\hat{v}_{s+1}} + \epsilon} \right)
$$
$$
- \sum_{s=0}^{N-1} \alpha_s \lambda \left( \int_0^1 \nabla_\theta f_{\theta_s(t)}(x) \, dt \right) \theta_s. \tag{5}
$$

Define

$$
\phi_s^{test}(x) := \int_0^1 \nabla_\theta f_{\theta_s(t)}(x) \, dt \in \mathbb{R}^{O \times D},
$$
$$
\phi_s^{train}(x_k) := \sum_{i=0}^{s} \alpha_{s,i} \, \mathbf{1}_{k \in B_i} \, w_{i,k} \, \frac{\nabla_\theta \big( L(f_{\theta_i}(x_k), y_k) \big)}{\sqrt{\hat{v}_{s+1}} + \epsilon} \in \mathbb{R}^D, \tag{6}
$$

and

$$\mathbf{r}_s := \alpha_s \lambda \theta_s \in \mathbb{R}^D.$$

With these definitions,

$$f_{\theta_N}(x) = f_{\theta_0}(x) - \sum_{k=1}^{M} \sum_{s=0}^{N-1} \phi_s^{test}(x) \phi_s^{train}(x_k) - \sum_{s=0}^{N-1} \phi_s^{test}(x) \mathbf{r}_s$$

which proves the claim. $\qquad\square$

**Remark.** The differentiability assumptions above are imposed for analytical convenience. They are not satisfied by all neural network architectures, e.g., networks with ReLU activations are not continuously differentiable with respect to the parameters at points where the pre-activation is exactly zero. However, these usually occur only on exceptional sets of parameter values and inputs. In our implementation and application of the above insights, we follow previous practice: One may assign suitable values at such points, so that the resulting training dynamics are still well defined. Thus, the theorem should be understood as applying either to continuously differentiable models or, more generally, to models for which these exceptional points are handled by an appropriate convention.

We further derive a similar result that aligns with the kernel formulation of (Bell et al., 2023).

**Corollary F.1** (Kernel decomposition for AdamW, in the style of (Bell et al., 2023)). *Under the assumptions of Theorem 3.1, define*

$$\phi_s^{train}(x_k) = \sum_{i=0}^{s} \alpha_{s,i} \, \mathbf{1}_{k \in B_i} \, w_{i,k} \left( \frac{\nabla_\theta f_{\theta_i}(x_k)}{\sqrt{\hat{v}_{s+1}} + \epsilon} \right)^\top \mathbf{a}_{i,k}.$$

*and*

$$\mathbf{a}_{i,k} := \left( \frac{\partial L(f_{\theta_i}(x_k), y_k)}{\partial f_{\theta_i}(x_k)} \right)^\top \in \mathbb{R}^O.$$

*Then, for every $x \in \mathcal{X}$,*

$$f_{\theta_N}(x) = f_{\theta_0}(x) - \sum_{k=1}^{M} \sum_{s=0}^{N-1} \phi_s^{test}(x) \left( \sum_{i=0}^{s} \alpha_{s,i} \, \mathbf{1}_{k \in B_i} \, w_{i,k} \left( \frac{\nabla_\theta f_{\theta_i}(x_k)}{\sqrt{\hat{v}_{s+1}} + \epsilon} \right)^\top \mathbf{a}_{i,k} \right) - \sum_{s=0}^{N-1} \phi_s^{test}(x) \mathbf{r}_s. \tag{7}$$

*Proof.* By Theorem 3.1,

$$\phi_s^{train}(x_k) = \sum_{i=0}^{s} \alpha_{s,i} \, \mathbf{1}_{k \in B_i} \, w_{i,k} \, \frac{\nabla_\theta\big(L(f_{\theta_i}(x_k), y_k)\big)}{\sqrt{\hat{v}_{s+1}} + \epsilon}.$$

Applying the samplewise chain rule gives

$$\nabla_\theta L(f_{\theta_i}(x_k), y_k) = \nabla_\theta f_{\theta_i}(x_k)^\top \mathbf{a}_{i,k}, \qquad \text{where } \mathbf{a}_{i,k} = \left( \frac{\partial L(f_{\theta_i}(x_k), y_k)}{\partial f_{\theta_i}(x_k)} \right)^\top.$$

Substituting this identity into the expression for $\phi_s^{train}(x_k)$ yields

$$\phi_s^{train}(x_k) = \sum_{i=0}^{s} \alpha_{s,i} \, \mathbf{1}_{k \in B_i} \, w_{i,k} \left( \frac{\nabla_\theta f_{\theta_i}(x_k)}{\sqrt{\hat{v}_{s+1}} + \epsilon} \right)^\top \mathbf{a}_{i,k}.$$

Substituting this expression into Theorem 3.1 proves (7). $\qquad\qquad\square$

**Remark.** In the setting of Theorem F.1, suppose that for each training sample $x_k$, the output-space loss gradient is constant across training steps, i.e.

$$\mathbf{a}_{i,k} = \mathbf{a}_k \qquad \text{for all } i = 0, \dots, N-1.$$

Then Equation 7 simplifies to

$$f_{\theta_N}(x) = f_{\theta_0}(x) - \sum_{k=1}^{M} \left( \sum_{s=0}^{N-1} \phi_s^{test}(x) \, \widetilde{\phi}_s^{train}(x_k)^\top \right) \mathbf{a}_k - \sum_{s=0}^{N-1} \phi_s^{test}(x) \mathbf{r}_s.$$

Thus, defining

$$I(x, x_k) := \sum_{s=0}^{N-1} \phi_s^{test}(x) \, \widetilde{\phi}_s^{train}(x_k)^\top \in \mathbb{R}^{O \times O},$$

we obtain

$$f_{\theta_N}(x) = f_{\theta_0}(x) - \sum_{k=1}^{M} I(x, x_k) \mathbf{a}_k - \sum_{s=0}^{N-1} \phi_s^{test}(x) \mathbf{r}_s.$$

This applies, for example, to the negative log-likelihood loss on log-probabilities. If the model outputs $f_{\theta_i}(x_k) \in \mathbb{R}^C$ are log-probabilities, e.g. via a final log-softmax layer, and the targets $y_k \in \mathbb{R}^C$ are one-hot encoded, then

$$L(f_{\theta_i}(x_k), y_k) = - \sum_{c=1}^{C} (y_k)_c \, (f_{\theta_i}(x_k))_c,$$

so that

$$\mathbf{a}_{i,k} = \left( \frac{\partial L(f_{\theta_i}(x_k), y_k)}{\partial f_{\theta_i}(x_k)} \right)^\top = -y_k,$$

which is independent of $i$.

**Remark.** Assume the setting of the previous remark, and in addition suppose that $f_{\theta_0}$ is constant. Then the resulting predictor admits the form of a regularized kernel machine in the sense of Bell et al. (2023): its prediction at $x$ is given by interactions with the training samples $x_k$ through the quantity

$$I(x, x_k) = \sum_{s=0}^{N-1} \phi_s^{test}(x) \, \widetilde{\phi}_s^{train}(x_k)^\top,$$

together with the additive weight-decay term

$$\sum_{s=0}^{N-1} \phi_s^{test}(x) \mathbf{r}_s.$$

To interpret $I$ as a kernel in the strict sense, one furthermore needs a unified conditional feature map that recovers the appropriate train-side and test-side features depending on the role of the input, analogous to the construction in Bell et al. (2023).

### F.2. Proof of Corollary 3.2

For the CNN model, we use gradient descent with momentum where the weight decay term contributes to the momentum buffer. We therefore derive the corresponding EPK decomposition also for this optimizer.

We first restate the statement.

**Statement** (Kernel Equivalence for GD with Momentum, repetition of Corollary 3.2 in the main text). *Let $f_\theta : \mathcal{X} \to \mathcal{Y}$ be a model with parameters $\theta \in \mathbb{R}^D$, and let*

$$\mathcal{D} = \{(x_1, y_1), \ldots, (x_M, y_M)\}$$

*be a dataset. Let*

$$L : \mathcal{Y} \times \mathcal{Y} \to \mathbb{R}_{\geq 0}$$

*be a per-sample loss. For each training step $i \in \{0, \ldots, N-1\}$, let $B_i \subseteq \{1, \ldots, M\}$ denote the mini-batch used at step $i$, and define the batch loss*

$$\mathcal{L}_i(\theta) := \sum_{k=1}^{M} \mathbf{1}_{k \in B_i} \, w_{i,k} \, L(f_\theta(x_k), y_k),$$

*where $w_{i,k} \in \mathbb{R}$ denotes the weight assigned to sample $(x_k, y_k)$ at step $i$.*

*Suppose the parameters are updated by gradient descent with momentum parameter $\beta \in [0, 1)$, learning rates $\alpha_s \in \mathbb{R}_{>0}$, and weight decay $\lambda \in \mathbb{R}_{\geq 0}$ according to*

$$\mathbf{b}_0 = 0, \qquad \mathbf{b}_{s+1} = \beta \mathbf{b}_s + \nabla_\theta \mathcal{L}_s(\theta_s) + \lambda \theta_s,$$

$$\theta_{s+1} = \theta_s - \alpha_s \mathbf{b}_{s+1}, \qquad s = 0, \ldots, N-1.$$

*Then, for every $x \in \mathcal{X}$,*

$$f_{\theta_N}(x) = f_{\theta_0}(x) - \sum_{k=1}^{M} \sum_{s=0}^{N-1} \phi_s^{test}(x) \, \phi_s^{train}(x_k) - \sum_{s=0}^{N-1} \phi_s^{test}(x) \, \mathbf{r}_s,$$

*where*

$$\theta_s(t) := \theta_s + t(\theta_{s+1} - \theta_s),$$

$$\phi_s^{test}(x) := \int_0^1 \nabla_\theta f_{\theta_s(t)}(x) \, dt \in \mathbb{R}^{O \times D},$$

$$\phi_s^{train}(x_k) := \sum_{i=0}^{s} \alpha_s \beta^{s-i} \mathbf{1}_{k \in B_i} \, w_{i,k} \, \nabla_\theta L(f_{\theta_i}(x_k), y_k) \in \mathbb{R}^D,$$

*and*

$$\mathbf{r}_s := \alpha_s \sum_{i=0}^{s} \beta^{s-i} \lambda \theta_i \in \mathbb{R}^D.$$

*Proof.* Unrolling the momentum recursion yields

$$\mathbf{b}_{s+1} = \sum_{i=0}^{s} \beta^{s-i} \left( \nabla_\theta \mathcal{L}_i(\theta_i) + \lambda \theta_i \right),$$

and therefore

$$\theta_{s+1} - \theta_s = -\sum_{i=0}^{s} \alpha_s \beta^{s-i} \nabla_\theta \mathcal{L}_i(\theta_i) - \alpha_s \sum_{i=0}^{s} \beta^{s-i} \lambda \theta_i.$$

Using

$$\nabla_\theta \mathcal{L}_i(\theta_i) = \sum_{k=1}^{M} \mathbf{1}_{k \in B_i} \, w_{i,k} \, \nabla_\theta L(f_{\theta_i}(x_k), y_k),$$

the rest follows exactly as in the proof of Theorem 3.1, yielding the claim with

$$\phi_s^{train}(x_k) = \sum_{i=0}^{s} \alpha_s \beta^{s-i} \mathbf{1}_{k \in B_i} w_{i,k} \nabla_\theta L(f_{\theta_i}(x_k), y_k)$$

and

$$\mathbf{r}_s = \alpha_s \sum_{i=0}^{s} \beta^{s-i} \lambda \theta_i.$$

$\square$

### F.3. Proof of Corollary 3.3

We restate Corollary 3.3 to which we outline the proof below:

**Statement** (Decomposition under differentiable post-composition). Assume the setup of Theorem 3.1 holds, and let $h : \mathbb{R}^O \to \mathbb{R}^P$ be differentiable. Define

$$\tilde{f}_\theta(x) := h(f_\theta(x)).$$

Then

$$\tilde{f}_{\theta_N}(x) = \tilde{f}_{\theta_0}(x) - \sum_{k=1}^{M} \sum_{s=0}^{N-1} \tilde{\phi}_s^{test}(x) \, \phi_s^{train}(x_k) - \sum_{s=0}^{N-1} \tilde{\phi}_s^{test}(x) \, \mathbf{r}_s,$$

where

$$\tilde{\phi}_s^{test}(x) := \int_0^1 \nabla h(f_{\theta_s(t)}(x)) \, \nabla_\theta f_{\theta_s(t)}(x) \, dt \in \mathbb{R}^{P \times D},$$

and $\phi_s^{train}(x_k)$ and $\mathbf{r}_s$ are the same train-side and regularization terms as in Theorem 3.1.

*Proof.* Let

$$\tilde{f}_\theta(x) := h(f_\theta(x)).$$

For each $s = 0, \ldots, N-1$, define

$$\theta_s(t) := \theta_s + t(\theta_{s+1} - \theta_s), \qquad t \in [0,1].$$

By the chain rule,

$$\nabla_\theta \tilde{f}_{\theta_s(t)}(x) = \nabla h(f_{\theta_s(t)}(x)) \, \nabla_\theta f_{\theta_s(t)}(x).$$

Thus, repeating the proof of Theorem 3.1 with $\tilde{f}_\theta$ in place of $f_\theta$, we obtain

$$\tilde{f}_{\theta_N}(x) = \tilde{f}_{\theta_0}(x) - \sum_{k=1}^{M} \sum_{s=0}^{N-1} \tilde{\phi}_s^{test}(x) \, \phi_s^{train}(x_k) - \sum_{s=0}^{N-1} \tilde{\phi}_s^{test}(x) \, \mathbf{r}_s,$$

where

$$\tilde{\phi}_s^{test}(x) = \int_0^1 \nabla_\theta \tilde{f}_{\theta_s(t)}(x) \, dt = \int_0^1 \nabla h(f_{\theta_s(t)}(x)) \, \nabla_\theta f_{\theta_s(t)}(x) \, dt \in \mathbb{R}^{P \times D}.$$

The train feature map $\phi_s^{train}(x_k)$ and the regularization term $\mathbf{r}_s$ are unchanged because they depend only on the optimizer trajectory in parameter space, not on the post-composition $h$. $\square$

As listed in the following remark, this extends ExPLAIND to decomposing and thus explaining a variety of other quantities describing model behavior.

**Remark.** Corollary 3.3 covers a range of quantities of interest obtained from the model output. For example:

1. **Loss values.** For fixed $y$, take
$$h_y(z) := L(z, y),$$
   provided $L(\cdot, y)$ is differentiable. Then the scalar quantity $L(f_{\theta_N}(x), y)$ admits the same decomposition.

2. **Single output coordinates.** For a class or coordinate $c$, take

$$h_c(z) := z_c.$$

   This yields a decomposition of the $c$-th logit, score, or output coordinate.

3. **Output margins.** For outputs $c \neq c'$, take
$$h_{c,c'}(z) := z_c - z_{c'}.$$

   This gives a decomposition of the margin between two classes.

4. **Linear projections.** For any matrix $A \in \mathbb{R}^{P \times O}$, take

$$h_A(z) := Az.$$

   This yields decompositions of arbitrary linear functionals or low-dimensional projections of the model output.

5. **Confidence-type quantities.** If the output parametrization permits it and the corresponding map is differentiable on the region of interest, one may also choose nonlinear functions such as entropy, logit gaps, or smoothed confidence scores.

We derive similar result for intermediate model activations thus completing proof of the claims in Theorem 3.3.

**Statement** (Decomposition of intermediate activations). Assume the setup of Theorem 3.1 holds, and suppose that the model admits a decomposition
$$f_\theta = h_\kappa \circ g_\iota, \quad \theta = (\iota, \kappa),$$

where $g_\iota : \mathcal{X} \to \mathbb{R}^P$ denotes an intermediate activation and $h_\kappa : \mathbb{R}^P \to \mathcal{Y}$ the remaining part of the model. Then the intermediate activations $g_\iota(x)$ admit an analogous decomposition:

$$g_{\iota_N}(x) = g_{\iota_0}(x) - \sum_{k=1}^{M} \sum_{s=0}^{N-1} \phi_{g,s}^{test}(x)\, \phi_s^{train}(x_k) - \sum_{s=0}^{N-1} \phi_{g,s}^{test}(x)\, \mathbf{r}_s,$$

where

$$\phi_{g,s}^{test}(x) := \int_0^1 \nabla_\theta g_{\iota_s(t)}(x)\, dt \in \mathbb{R}^{P \times D},$$

and $\phi_s^{train}(x_k)$, $\mathbf{r}_s$ are as in Theorem 3.1.

*Proof.* Apply Theorem 3.1 to the map
$$\tilde{f}_\theta(x) := g_\iota(x),$$

viewed as the output of the submodel determined by the parameters $\theta = (\iota, \kappa)$. The optimizer trajectory in parameter space is unchanged, so the train-side quantity $\phi_s^{train}(x_k)$ and the regularization term $\mathbf{r}_s$ remain the same. Only the test-side feature map changes, becoming

$$\phi_{g,s}^{test}(x) = \int_0^1 \nabla_\theta g_{\iota_s(t)}(x)\, dt.$$

Substituting this into the decomposition from Theorem 3.1 yields the claim. $\qquad \square$

### F.4. Strategies for scaling to larger settings

First we prove the corollary according to which we can accumulate gradients in the train feature maps before computing the dot product of the feature maps.

**Corollary F.2** (Partitionwise accumulation of train feature maps). *Let $I_1, \ldots, I_P$ be a partition of $\{1, \ldots, M\}$. For each $s \in \{0, \ldots, N-1\}$ and $p \in \{1, \ldots, P\}$, define*

$$\phi_s^{train}(I_p) := \sum_{k \in I_p} \phi_s^{train}(x_k).$$

*Then, for every $x \in \mathcal{X}$,*

$$f_{\theta_N}(x) = f_{\theta_0}(x) - \sum_{p=1}^{P} \sum_{s=0}^{N-1} \phi_s^{test}(x) \, \phi_s^{train}(I_p) - \sum_{s=0}^{N-1} \phi_s^{test}(x) \, \mathbf{r}_s.$$

*In particular, if only aggregate influences over the $I_p$ are required, the train feature maps may be accumulated partition-wise before taking the dot product with the test feature map.*

*Proof.* Since $I_1, \ldots, I_P$ form a partition of $\{1, \ldots, M\}$, we may rewrite the sum over training samples in Theorem 3.1 as

$$\sum_{k=1}^{M} \sum_{s=0}^{N-1} \phi_s^{test}(x) \, \phi_s^{train}(x_k) = \sum_{p=1}^{P} \sum_{s=0}^{N-1} \sum_{k \in I_p} \phi_s^{test}(x) \, \phi_s^{train}(x_k).$$

By linearity of matrix-vector multiplication,

$$\sum_{k \in I_p} \phi_s^{test}(x) \, \phi_s^{train}(x_k) = \phi_s^{test}(x) \sum_{k \in I_p} \phi_s^{train}(x_k) = \phi_s^{test}(x) \, \phi_s^{train}(I_p).$$

Substituting this into the decomposition from Theorem 3.1 yields

$$f_{\theta_N}(x) = f_{\theta_0}(x) - \sum_{p=1}^{P} \sum_{s=0}^{N-1} \phi_s^{test}(x) \, \phi_s^{train}(I_p) - \sum_{s=0}^{N-1} \phi_s^{test}(x) \, \mathbf{r}_s,$$

which proves the claim. $\square$

Next, we remark that under subsampling the training steps, one has to decouple influences of the first moment term and the actual training batch, as for this setting only a flattened version of the first moment is available through the optimizer checkpoint.

*Remark* F.3 (Implementation under checkpoint subsampling). In practical implementations, the decomposition in Theorem 3.1 may be evaluated only at a subsampled set of checkpoints rather than at every optimization step. In this case, the contribution of the data through the Adam first-moment term is not fully recoverable at the level of individual training examples, since the first moment aggregates past gradients between two stored checkpoints.

This has two consequences. First, although the example-wise decomposition of the first-moment contribution is unavailable, its aggregate effect on the parameters is still accessible through the optimizer state, since the unexpanded first-moment average is stored explicitly. Thus, one may still quantify how much of the model behavior is introduced by the first moment through parameters and training, even if this contribution cannot be attributed exactly to individual examples.

Second, the data-dependent contribution of a training batch through the first moment can still be approximated from its explicit gradient contribution at the step where it appears. Indeed, since the Adam first moment is an exponential moving average, a gradient contribution introduced at step $i$ reappears in later optimizer states with geometrically decaying weights. More precisely, if a batch contributes a gradient term $g_i$ at step $i$, then its contribution to the first moment at a later step $s \geq i$ is weighted by a factor proportional to $\beta_1^{s-i}$. Hence, over an interval of steps, its cumulative contribution through the first moment can be extrapolated by scaling its original contribution by the corresponding geometric sum. In particular, if one only observes checkpoints at a coarser temporal resolution, the omitted first-moment influence of an example or batch may be estimated by multiplying its explicit contribution at occurrence time with the sum of the $\beta_1$-weights with which it persists across the skipped steps. Note that we did not consider this rescaling in our EuroLLM experiments.

### F.5. Influence accumulation

The ExPLAIND formulation of influence gives us a tensor that enables the attribution of model behavior to each of these dimensions across different, unified perspectives—these are the dimensions of influence we sum over—and granularities, corresponding to the size of the sets we sum over. Here we expand the examples given in the main text:

- **Single parameter.** The influence of a single parameter $\theta^{(i)}$ on a given prediction of a sample $x$, is given by the accumulation of parameter-wise kernel scores over the training set , i.e.

$$\Psi(\theta^{(i)}, x)_j = \sum_{k=1}^{M} \sum_{s=0}^{N-1} \sum_{i=1}^{D} \psi_s(\theta^{(i)}, x, x_k)_j.$$

- **Layer-level at a specific training step.** The influence of all parameters in a layer $\Theta_L$ at training step $s$ on the prediction for $x$ is

$$\Psi_s(\Theta_L, x) = \sum_{\theta^{(i)} \in \Theta_L} \sum_{k=1}^{M} \sum_{j=1}^{O} \psi_s(\theta^{(i)}, x_k, x)_j.$$

- **Data partition through a layer.** The influence of a layer $\Theta_L$ on $x$ due to a subset of training data $X \subseteq \mathcal{D}_{train}$ (for example, a data class) at step $s$ is

$$\Psi_s(\Theta_L, x, X) = \sum_{\theta^{(i)} \in \Theta_L} \sum_{x_k \in X} \sum_{j=1}^{O} \psi_s(\theta^{(i)}, x_k, x)_j.$$

# G. Additional experiments, results and figures

In this appendix, we provide additional figures from our experiments that were omitted from the main text as well as the data attribution experiment.

## G.1. Causal data attribution experiment

We evaluate the vanilla ExPLAIND influence scores accumulated to the data level in terms of their ability to linearly predict the effect of changes in the training data. We do this in terms of the linear data modeling score (LDS; Ilyas et al. (2022); Park et al. (2023)) which is a commonly used metric for evaluating non-unified data attribution methods. Intuitively, LDS is the correlation of the predictions of a model re-trained on a subset of data and predictions derived linearly from the influence scores of the sane subset.

Note that the influence scores of our framework are not designed to provide such counterfactual estimations. Rather, ExPLAIND quantifies the exact influences of the training data on a given model instance. In other words, while LDS evaluates the influence quantification of data on the *average* training run, our scores are *specific* run.

**Methodology.**  Nevertheless, we hypothesize that ExPLAIND might perform non-trivially in terms of LDS. In line with our framework, for a training sample $x_k$, prediction $x$ and output $j$, we directly use our influence scores $\Phi(\mathcal{S}, \Theta, x_k, x)_j$ accumulated over the training steps $\mathcal{S}$ and model parameters $\Theta$. For a subset of training samples $X'$, we follow Park et al. (2023) and predict the prediction of $x$ at output $j$ of a model trained on training data $X'$ as $\Phi(\mathcal{S}, \Theta, X', x)_j$.

**Experiment settings.**  To compute LDS, we retrain the MNIST model on 1000 random subsets of the training data for fractions $\alpha = \{0.5, 0.7, 0.9, 0.95, 0.99, 0.999\}$, where the 0.999 setting refers to the leave-one-out (LOO) setting. We compare against single-model TRAK (Park et al., 2023) and TracIn (Pruthi et al., 2020) as baselines. Furthermore, we compute the LDS for the CIFAR model for $\alpha = 0.5$, because retraining and attribution is more expensive there due to the larger data and model size. All hyperparameters related to training, architecture, etc. are the same as in the MNIST run presented in the main paper (see Section E.1).

**Results.**  In the CIFAR setting, we find that ExPLAIND scores accumulated to the data level (LDS=0.06) perform comparable to the TracIn and TRAK baselines in the CIFAR setting (both LDS=0.05).

We plot the LDS across the data sparsity levels in the MNIST setting in Figure 7. There, ExPLAIND generally performs close to the TracIn baseline, but both are outperformed by TRAK in all but the leave-one-out ($\alpha = 0.999$) setting, where ExPLAIND scores the highest.

Further, we report that the wall-clock time to compute the ExPLAIND data attribution scores ($170s$) in the MNIST setting is comparable to those of TRAK ($114s$) and TracIn ($100s$), i.e. in comparable settings, the runtime of ExPLAIND is equivalent to that of other methods. This further shows that the complexity of ExPLAIND is manageable in practice and comparable even to non-unified baselines.

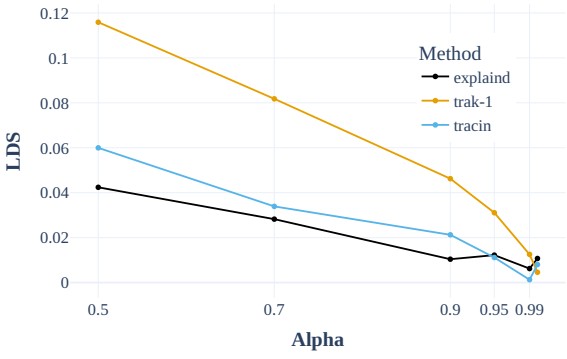

*Figure 7.* **LDS data attribution.** We plot the linear datamodeling score of ExPLAIND across different levels of data sparsity on out MNIST setting and compare against two popular baselines, single-model TRAK and TracIn.

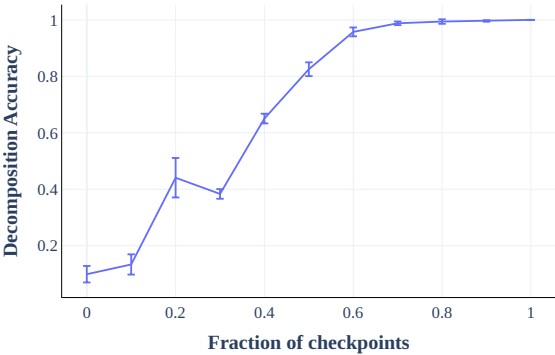

*Figure 8.* **Sensitivity analysis of accumulated scores.** To test the effect of subsampling the training steps considered for computing the ExPLAIND decomposition, we perform a sensitivity analysis. As a simple heuristic to identify promising steps, we rank the steps according to the overall test loss decrease, cut off at at different fractions of parameters (x axis), and plot the accuracy of the predictions resulting from summing up the remaining scores, in other words the agreement with the actual model's predictions (y axis). We find that this naive way of subsampling the training steps is accurate up to around $40\%$ of all steps pruned.

### G.2. Sensitivity of the decomposition to subsampling training steps

Subsampling the number of considered training steps for the decomposition is a promising direction for speeding up the runtimes of ExPLAIND. Optimally, we would hope that such a sampled decomposition would still recover the original model's behavior by summing up the scores like in Theorem 3.1. Note, however, that for gaining local insights into the training dynamics of models, this condition does not necessarily need to be met: Decomposing a single step faithfully comes with we same mathematical foundation as decomposing an entire training run, though with a different, more local scope, i.e. an explanation based only on a slice of the tensor of influences.

Up to which degree of sparsification of the training trajectory can we still recover an accurate global decomposition? We study this question in the MNIST setting with a simple experiment. To obtain a more realistic and longer training trajectory, we increase the number of epochs to five, while keeping the remaining hyperparameters fixed as described in Section E.1. We rank the update steps by the absolute change they induce in the loss on a held-out batch of samples.

For different values of $X$, we then restrict the decomposition to the subset $\mathcal{S}_{\mathrm{top}X} \subseteq \{0, \dots, N-1\}$ consisting of the top $X\%$ of steps with largest held-out loss change, and compare the resulting reconstructed predictions to the actual model predictions as in Section 3.1. Concretely, the sparse decomposition is given by

$$f'_{\theta_N}(x)_j := f_{\theta_0}(x)_j - \sum_{k=1}^{M} \sum_{s \in \mathcal{S}_{\mathrm{top}X}} \sum_{i=1}^{D} \psi(s, \theta^{(i)}, x_k, x)_j - \sum_{s \in \mathcal{S}_{\mathrm{top}X}} \sum_{i=1}^{D} \psi^{reg}(s, \theta^{(i)}, x)_j.$$

As shown in Figure 8, we find that the reconstructed prediction remains close to the true model prediction already when retaining only the top $60\%$ of training steps, that is, after discarding the remaining $40\%$ of steps. In other words, for our MNIST model, the number of steps included in the decomposition can be reduced by almost half while still retaining an accurate reconstruction.

Taken together, these findings suggest that the training trajectory contains substantial redundancy from the perspective of decomposition, but that naive step subsampling exposes an inherent fidelity-runtime trade-off. The heuristic used here is intentionally simple, and future work should investigate whether adaptive or influence-aware step-selection schemes can compress the trajectory more aggressively while preserving the accuracy of the resulting decomposition.

## G.3. Additional figures

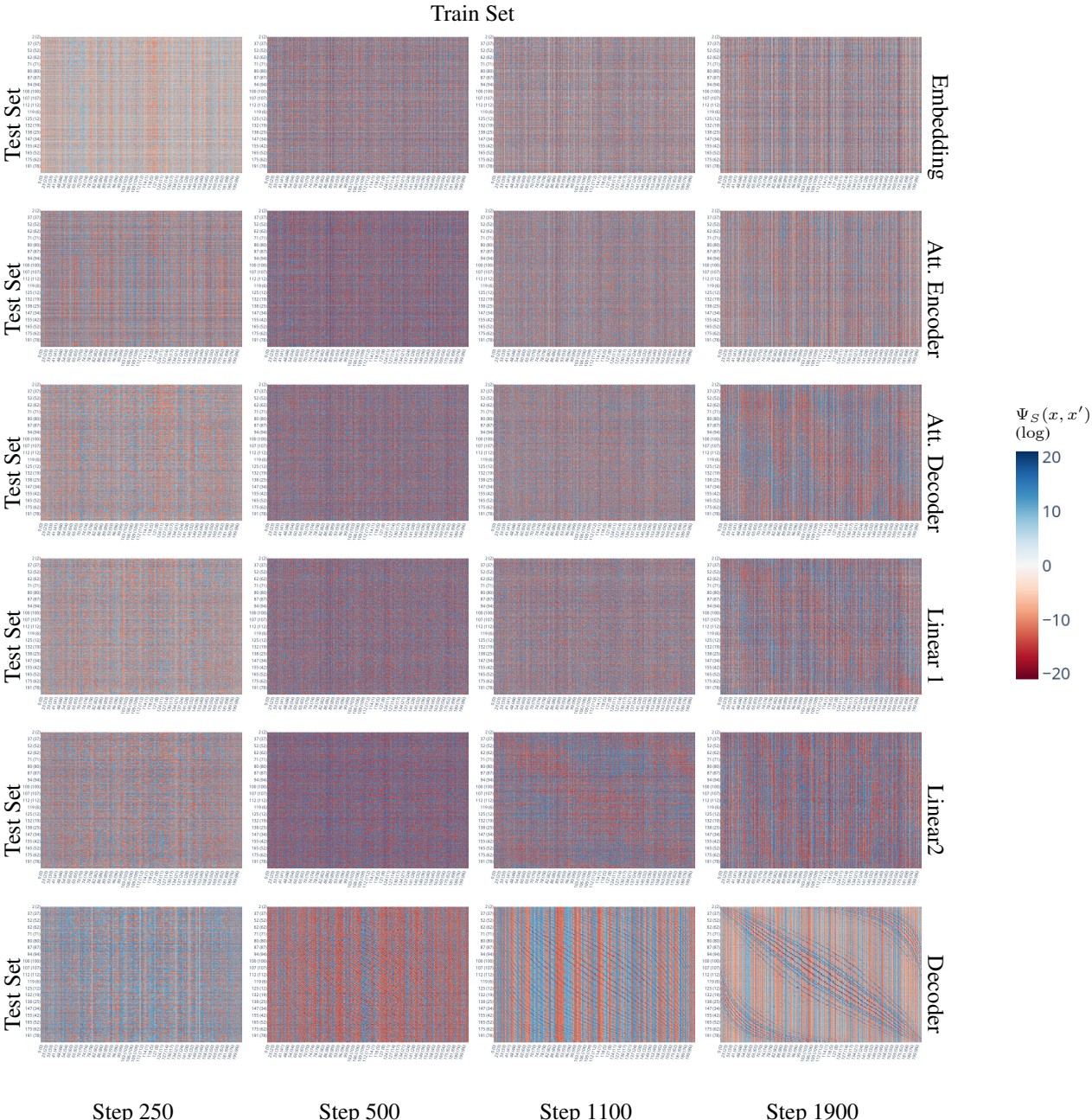

*Figure 9.* **Other slices of the scores of the Transformer model.** We plot the accumulated $\Psi_S(\Theta_{layer}, x, x')$ of all layers $\Theta_{layer}$ for predictions of the test set $x'$ and the training set $x$ accumulated over preceding 50 steps $S$, labeled with sum of inputs $a + b$ and respective result $(a + b \mod c)$.

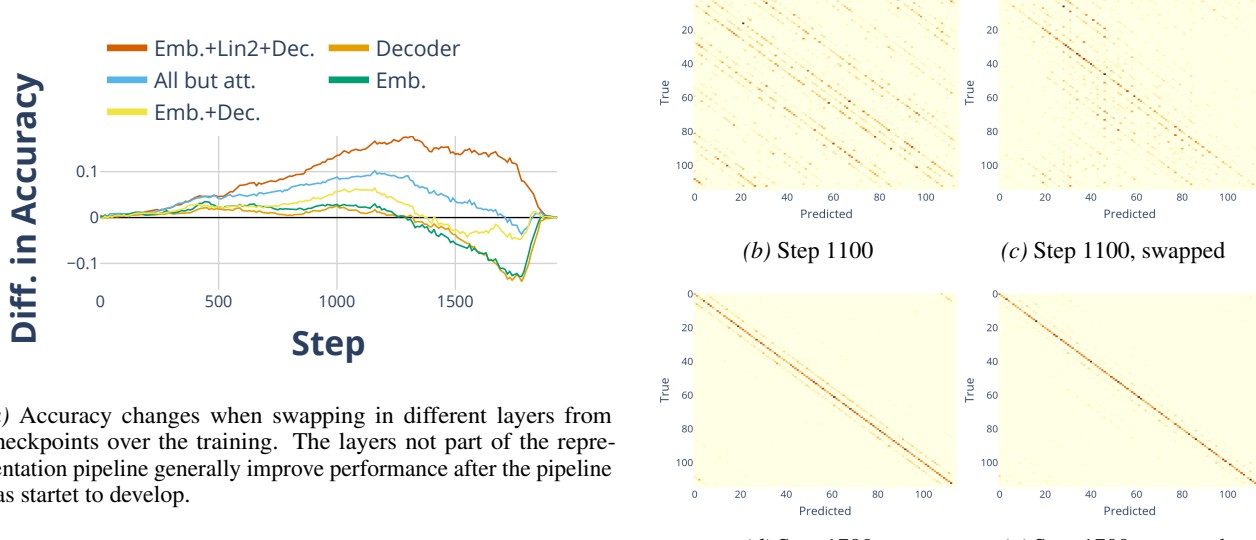

*(a)* Accuracy changes when swapping in different layers from checkpoints over the training. The layers not part of the representation pipeline generally improve performance after the pipeline has startet to develop.

*(b)* Step 1100

*(c)* Step 1100, swapped

*(d)* Step 1700

*(e)* Step 1700, swapped

*Figure 10.* **Layer swapping validations. Left:** We swap different layers of the final Transformer model into checkpoints across the training trajectory and find that the layers involved in the final alignment phase (the embedding, second linear layer and the decoder), improve accuracy by over $15\%$, supporting our hypothesis of a pipeline of intermediate layers developing a generalizing representation before the final Grokking phase. **Right:** Confusion matrices of two unedited checkpoints and their respective swapped versions. Note the decrease in systematic errors on the off-diagonals.

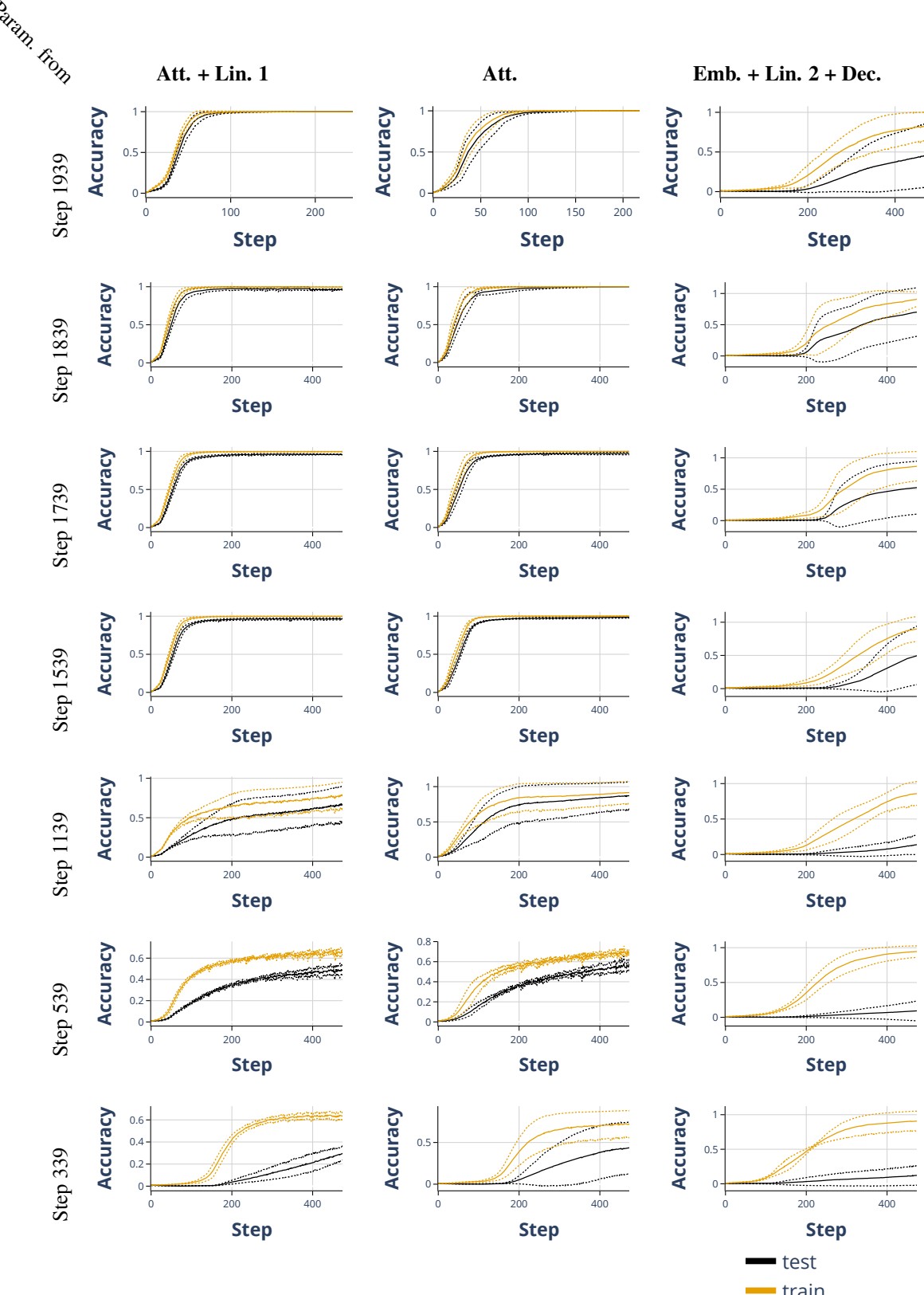

*Figure 11.* **Training on grokked intermediate representation pipelines.** We train a model initialized with the different parameters taken from different checkpoints and model components and initialize the rest at random. This leads to rapid, and direct generalization over 5 different runs when we take the attention weights (here 'Att.' refers to both the encoder and decoder of the attention layer) from later training steps, when the intermediate pipeline has already generalized. We report the mean over five runs and standard deviation as dotted error bars.

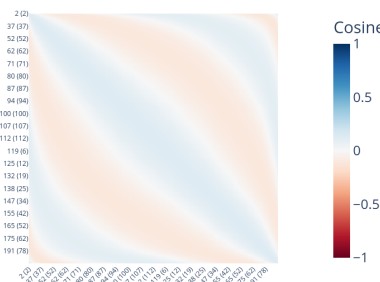

*Figure 12.* **Lasso regression on influence similarities.** We fit the second linear layers similarity domain introduced in steps 1850 to 1900 with a lasso regression. Features are the cosines and sines of frequencies from 2 to 113 of the pairwise differences of the sums of the samples. Shown: Predictions of the similarity as predicted by the regression. The resulting similarity pattern indicates that the model indeed learns to map the samples into space where distance is approximately a cosine of frequency 113. We report the exact regression coefficients in Table 2.

*Table 2.* **Lasso regression on influence similarities.** We fit the second linear layers similarity domain introduced in steps 1850 to 1900 with a lasso regression. Features are the cosines and sines of frequencies from 2 to 113 of the pairwise differences of the sums of the samples. The table shows all non-zero regressions coefficients of cosines frequency.

| cos frequency | Regression Coefficient |
| --- | --- |
| 113 | 0.112333 |
| 76 | 0.008232 |
| 51 | 0.004451 |
| 75 | 0.003616 |
| 52 | 0.001763 |
| 37 | 0.000679 |
| 13 | 0.000483 |
| 28 | 0.000259 |
| 38 | 0.000093 |

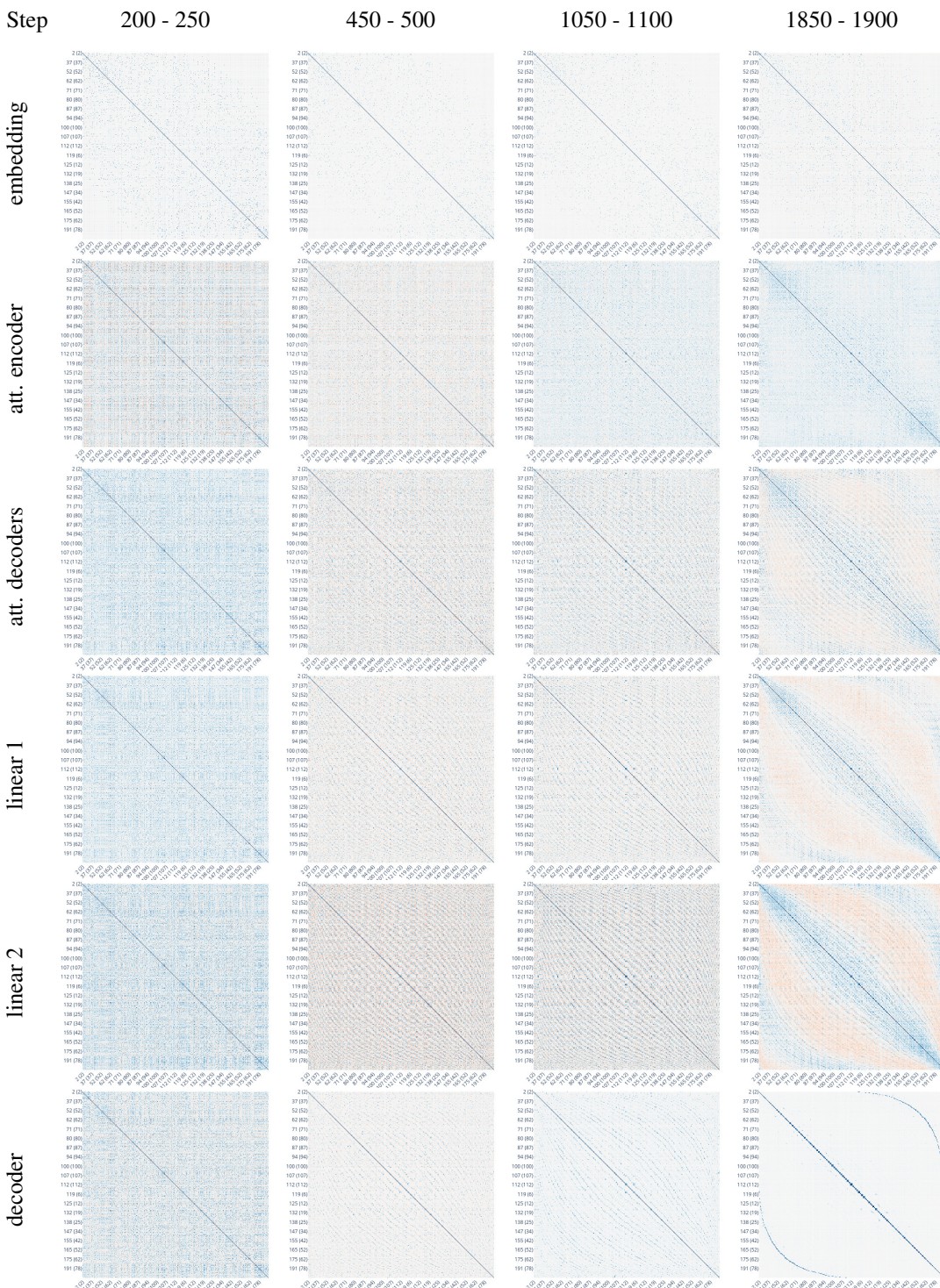

*Figure 13.* **Full similarity plots.** Similarity plots of test set predictions of the Transformer model accumulated over different training stages.

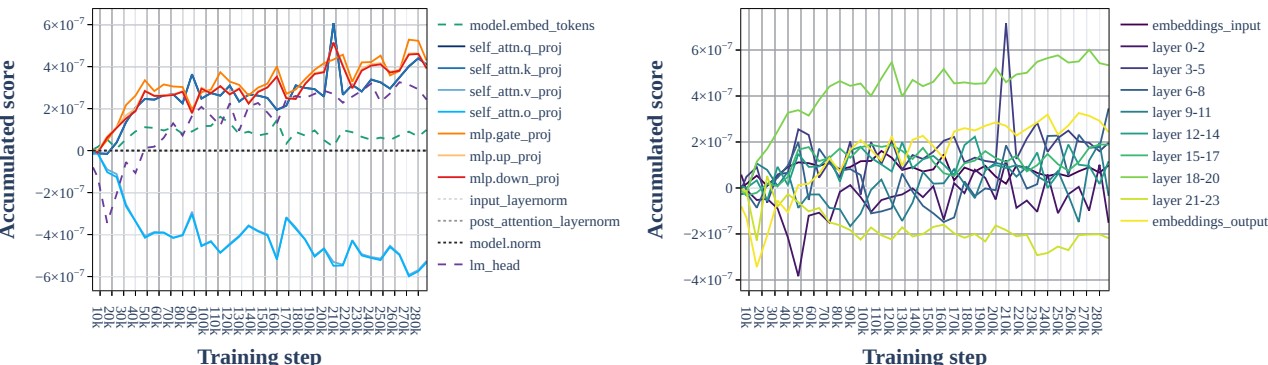

*Figure 14.* **EuroLLM regularization influences.** Accumulated regularization influence scores $\Phi^{reg}(s, \Theta)$ based on the decomposition of the loss trajectory of EuroLLM pretraining by parameter type (left) and layer depth (right). Mirroring the phase transition at 60K observed in the data influences, the regularization influences shift in the output embeddings (lm_head) from being loss-decreasing (i.e. helpful) to loss-increasing in the second phase. Other trends are less pronounced.

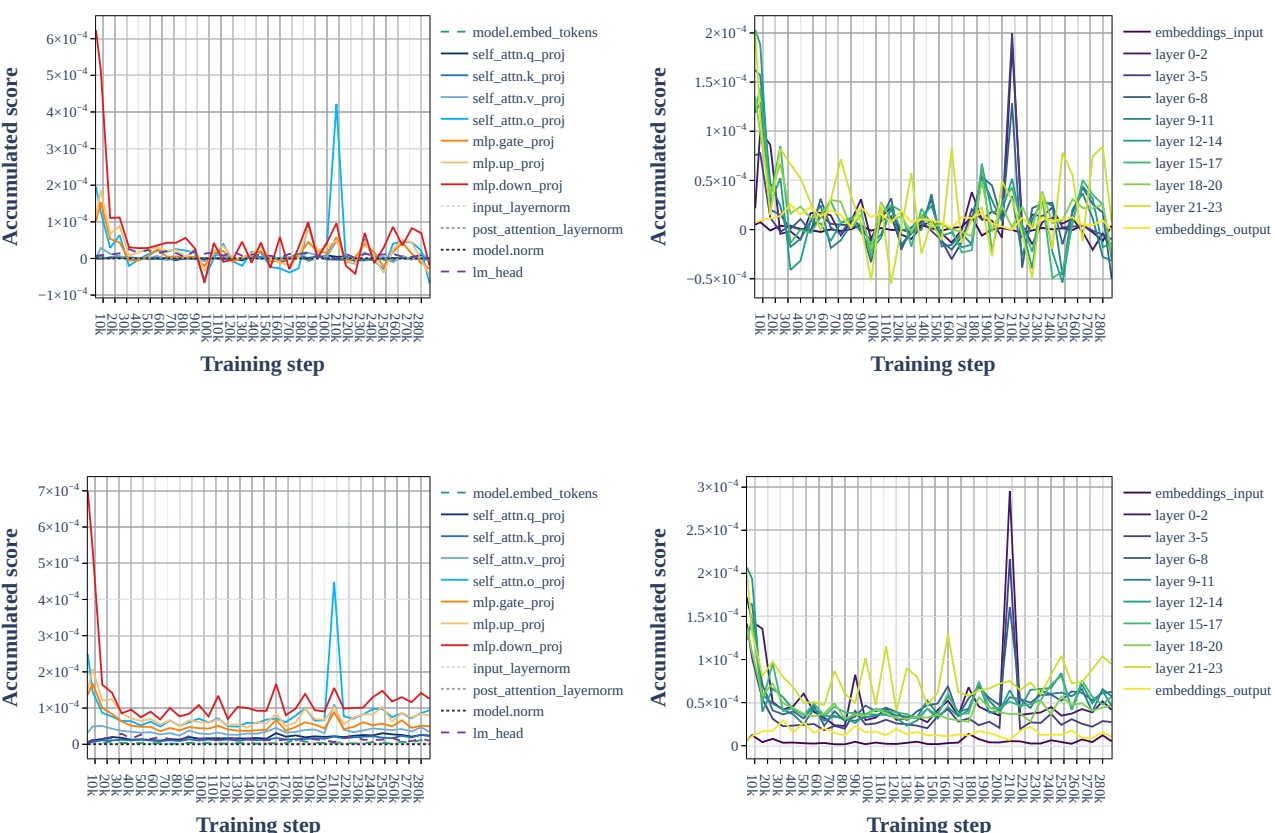

*Figure 15.* **EuroLLM first moment influences.** Accumulated first moment influence scores $\Phi^{fm}(s, \Theta)$ based on the decomposition of the loss trajectory of EuroLLM pretraining by parameter type (left) and layer depth (right). We plot both, the signed (top) and absolute scores (bottom) to visualize with and without the oscillatory nature of the scores.

