# OpenReview forum: "ExPLAIND: Unifying Model, Data, and Training Attribution to Study Model Behavior"
_ICML.cc/2026/Conference — ICML 2026 regular_

### Official Review · Reviewer_Ga8a · 2026-03-11

**Soundness:** 3
**Presentation:** 3
**Significance:** 3
**Originality:** 3
**Overall Recommendation:** 5
**Confidence:** 3

**Summary:**

The paper introduces **ExPLAIND**, a framework for analyzing neural network behavior by jointly attributing model predictions to **training data, model parameters, and optimization steps**. The central idea is to provide a unified view of attribution by extending the notion of gradient path kernels, enabling a decomposition of model predictions into contributions associated with different elements of the training process.

The authors derive attribution scores that quantify how individual parameters, training examples, and optimization steps influence a model’s predictions. This formulation connects several previously studied perspectives on model interpretability such as parameter importance, data influence, and training dynamics within a single theoretical framework. The proposed approach provides feature maps associated with the gradient path kernel, which allow the computation of influence scores across these different dimensions.

The paper further demonstrates how these attribution scores can be used to study model behavior in practice. In particular, the authors show applications including parameter pruning and the analysis of training dynamics. They also present experiments illustrating how the proposed framework can approximate the behavior of trained neural networks and use it to analyze phenomena such as grokking in transformer models.

Overall, the paper proposes a unified theoretical framework for attribution across model parameters, training data, and training trajectory, and demonstrates its utility through several empirical analyses of neural network behavior.

**Compliance With Llm Reviewing Policy:**

Affirmed.

**Key Questions For Authors:**

Scalability of the proposed framework. The proposed attribution scores rely on quantities derived from gradient path kernels. Could the authors clarify the computational cost of computing these scores in practice? In particular, how does the approach scale with model size, number of training steps, and dataset size?

Approximation accuracy. The paper suggests that the proposed formulation can approximate the behavior of trained neural networks. Could the authors provide additional quantitative evaluation of the approximation quality, for example by reporting prediction errors or correlations between the original model outputs and the kernel-based reconstruction?

Comparison with existing influence methods. Several prior methods analyze training data or optimization influence (e.g., influence functions or trajectory-based methods). Could the authors clarify how the proposed framework compares to these approaches in terms of computational cost, interpretability of the resulting scores, and empirical performance?

Sensitivity and robustness. How sensitive are the attribution scores to training details such as learning rate schedules, optimizer choice, or random initialization? Understanding this could help clarify the robustness of the insights derived from the method.

Practical usage in large models. The paper demonstrates applications on several neural network architectures. Could the authors discuss whether the proposed method can be applied to larger modern models (e.g., larger transformers), and what practical limitations might arise?

Insights from the grokking analysis. In the grokking case study, could the authors clarify what new insights the proposed framework provides compared to existing analyses of this phenomenon?

**Limitations:**

Yes. The paper discusses limitations of the proposed framework and focuses on analytical insights rather than deployment in real-world systems. No significant direct societal risks appear to arise from the proposed methodology.

**Strengths And Weaknesses:**

Strengths

• **Unified perspective on attribution.** The paper proposes a unified framework that connects attribution to training data, model parameters, and optimization trajectory. This perspective is conceptually appealing and helps bridge several previously separate lines of work in interpretability and influence analysis.

• **Theoretical formulation based on gradient path kernels.** The work extends the notion of gradient path kernels and derives feature maps that enable attribution scores across different components of the training process. This provides a principled formulation for analyzing how model behavior emerges during training.

• **Potential practical applications.** The framework enables several analyses of model behavior, including parameter importance and pruning, as well as studying training dynamics. These use cases illustrate how the proposed attribution scores can provide insights into model mechanisms.

• **Empirical demonstrations.** The paper presents experiments on neural network models (including CNNs and transformers) showing that the proposed formulation can approximate trained models and can be used to analyze phenomena such as grokking.

• **Clarity of motivation.** The paper clearly motivates the need for a unified approach to attribution across different elements of the learning process, which is a relevant problem for interpretability and understanding model behavior.

Weaknesses

• **Scalability and computational cost.** The paper does not fully clarify the computational cost of computing the proposed attribution scores in realistic training settings. Since the method relies on gradient-path-based quantities, it would be helpful to better understand how the approach scales with model size, dataset size, and training duration.

• **Limited empirical validation.** While the paper provides several illustrative experiments, the empirical evaluation appears somewhat limited in scope. Additional experiments across more architectures, datasets, or tasks would strengthen the evidence that the framework is broadly useful in practice.

• **Comparison with existing influence methods.** The paper would benefit from clearer comparisons with existing approaches for data or training attribution (e.g., influence functions or trajectory-based methods). It would help clarify the practical advantages of the proposed framework.

• **Clarity of practical usage.** While the theoretical framework is well motivated, it is not always fully clear how practitioners should apply the method in large-scale settings or how sensitive the results are to implementation choices.

• **Scope of contributions.** The paper addresses several different aspects (theoretical formulation, attribution analysis, pruning, and training dynamics analysis). While this demonstrates the flexibility of the framework, the paper could benefit from a more focused empirical validation of the most central contribution.

---

> ### Author Rebuttal · Authors · 2026-03-31
>
> Thank you for your encouraging feedback. We reply as follows:
>
> > clarify the computational cost
>
> Since computational cost is indeed a factor that can limit our approach, we included a discussion on computational complexity (see section 4.1 Efficiency), as well as a case study implementing our proposed strategies to scale to large models (see Discussion/Larger settings.), where we also discuss the actual runtime of this approach (see line 437). We also detail the compute for the other experiments in Appendix E.3
>
> > the empirical evaluation appears somewhat limited in scope
>
> We agree that further settings would be an interesting addition to the paper. We thus include a new CNN model we train on MNIST. Agreeing with our other three settings, we find the predictions of this model to be reproduced by the ExPLAIND decomposition (EPK Accuracy 100% and KL Divergence of$ 5.7\cdot10^{-8}$). Note also our new empirical settings described in the other reviews.
> Our main goal is to introduce, validate and showcase an interpretability framework. Therefore, we wanted to dedicate enough space to studying a model and showing some of the new forms of explanation our framework provides.
>
> > clearer comparisons with existing approaches for data or training attribution
>
> We agree that there are interesting baselines for data attribution. As we argue in lines 420-429, a comparison is difficult as ExPLAIND attributes data for a specific model instance, whereas other approaches quantify the average influence of data over many models. Despite this, we introduce two additional evaluations of the CIFAR and the new MNIST setting in terms of the linear datamodeling score (LDS; Ilyas et al., 2022). We find that ExPLAIND scores accumulated to the data level perform comparable to the TracIn and TRAK baselines in the former setting, while in the latter TracIn and ExPLAIND are outperformed by TRAK. For details, please check our response to Reviewer Hbya, point 1 due to the character limit.
>
> > how sensitive the results are to implementation choices [or] training details?
>
> Our mathematical derivations imply that ExPLAINDs influence scores are accurate for any gradient descent setting given an accurate enough estimation of the integral in the test feature map. We validate ExPLAIND over four diverse settings which all show our scores to be highly accurate at the 100 integral step threshold. The paper thus establishes reasonable empirical evidence to support this.
>
> Regarding sensitivity of the implementation of ExPLAIND, we present two new results quantifying the sensitivity of our two efficiency strategies in the new MNIST model. For details, please check our response to Reviewer Hbya, point 2 due to the character limit.
>
> > how practitioners should apply the method in large-scale settings
>
> Utilizing the additional page of the camera ready paper, we extend the case study of the large-scale EuroLLM with more discussion on how ExPLAIND can be applied in realistic settings (see first point of our comment to Reviewer is1k for details) as well as two new results, plotting layer-wise influences over time for this model. This paints a much clearer picture of how ExPLAIND can help in realistic settings and what its use-case is compared to other methods.
>
> > more focused empirical validation of the most central contribution.
>
> We are not sure which further experiments could have helped. Could you please elaborate? We would be very happy to address your concerns. Please note our new data attribution validation experiments, new MNIST setting, and sensitivity study we have run to address your and the other reviewers’ feedback, extending our already extensive quantitative experiments.
>
> > additional quantitative evaluation of the approximation quality
>
> We refer to the results in Table 1 and our case study into EuroLLM (line 406 onwards), which both show our scores to reproduce model behavior exactly. Both, in terms of accuracy (i.e. agreement of argmaxes) and KL-divergence (i.e. similarity of entire output distribution). Table 1 will further be extended with the MNIST model results, which also support these findings.
>
> > discuss whether the proposed method can be applied to larger modern models
>
> We discuss this in Section 4.1 as well as Discussion/Larger settings, which we further extend with more discussion and illustrative new results on EuroLLM (see above). We also add new results and discussion addressing efficiency and applicability of our approach (see above).
>
> > In the grokking case study[...]clarify what new insights
>
> Our work refines Nanda et al. (2023)’s proposed learning phases (line 231 to 235 and 318 to 320 and 352 to 354), reinterpreting the last phase as one of alignment of predictions with an intermediate representation pipeline formed in the second phase, and Huang et al. (2024)’s efficiency perspective (lines 320 to 323). We further attribute relevant works in the Appendix (lines 670 to 677).

---

> > ### Author Rebuttal · Reviewer_Ga8a · 2026-04-02
> >
> > The rebuttal addresses several of my concerns and improves the overall clarity of the submission. In particular, the authors provide additional empirical results (including a new MNIST setting), as well as new comparisons with prior attribution methods (e.g., TracIn, TRAK, and LDS), which help better position the contribution. The added discussion on scalability and large-scale applications (e.g., EuroLLM) is also helpful in clarifying the intended use cases of the method.
> >
> > While some limitations remain particularly regarding scalability to very large models and the still relatively limited empirical validation—the rebuttal provides sufficient additional evidence and clarification to support the core claims of the paper.
> >
> > Overall, I find the proposed framework to be an interesting and conceptually valuable contribution that unifies multiple perspectives on attribution. Given the improvements in the rebuttal and the potential of the approach, I am in favor of acceptance.

---

### Official Review · Reviewer_is1k · 2026-03-13

**Soundness:** 3
**Presentation:** 4
**Significance:** 4
**Originality:** 3
**Overall Recommendation:** 5
**Confidence:** 3

**Summary:**

The paper presents ExPLAIND, a unified framework that integrates interpretability of model parameters, data and training steps. ExPLAIND is based on the Exact Path Kernel (EPK) but it extends it to more modern training regimes. After theoretically establishing their method the authors validate it by showing that the derived importance scores can serve as useful pruning scores. The authors then use ExPLAIND to study a model known to experience Grokking and find a novel alignment phase. Lastly they show how their method can even be applied to study models with over a billion parameters.

**Compliance With Llm Reviewing Policy:**

Affirmed.

**Final Justification:**

I choose to main my score of 5. This is a good paper but I think further empirical comparisons with existing methods would be beneficial and I am too unfamiliar with the domain to raise my score to a 6.

**Key Questions For Authors:**

See weaknesses

**Limitations:**

Yes.

**Strengths And Weaknesses:**

### Strengths
- Integrating explanations of model outputs across training examples, parameter and training steps is an incredibly valuable contribution.
- The paper is well written, with clear explanations and experiments.
- The method is well justified theoretically and empirically. I appreciate the fact that the authors actually tried using ExPLAIND on large scale models.

### Weaknesses
- I would appreciate a discussion of this models applicability in practice, what type of results can be obtained with such methods beyond the artificial grokking scenario? Of similar importance are the limitations of this method, where does it fail?
- There is currently no comparison of ExPLAIND with other similar methods, this would be very informative even if, as the authors state, other methods aren't unified across the three dimensions.
- When discussing the 1.7B model results

I would really like to see the authors open-source the code and hopefully even build a library that would allow the application of ExPLAIND in a variety of settings with a variety of models. I believe this would bring tremendous value to the research community.

---

> ### Author Rebuttal · Authors · 2026-03-31
>
> Thank you for your thoughtful and encouraging review. Regarding the weaknesses you point out, we answer as follows:
>
> > I would appreciate a discussion of this models applicability in practice
>
> We agree that further discussion of ExPLAINDs applicability in more realistic settings would strengthen our paper. As we outline in the paper, accumulating over the tensor of influences can yield a lot of different types of explanations (see Grokking study) which can be obtained in the large settings, though perhaps over a subset of the training steps and larger partitions of the data (e.g. documents instead of single tokens for an LM). This does not break the theoretical guarantees of our framework: The resulting scores are still exact for their respective parts of the tensor (see Discussion/Larger settings.).
>
> To address this valid critique, we will dedicate the extra page in the camera ready version to extend our EuroLLM case study. There, we will discuss the explanations one can still expect to compute in the large setting and present a layer-wise attribution result similar to the one presented in Fig. 3(a) of the Grokking study. As suggested by the ICML review guidelines, we attach both plots where we group the influences of EuroLLM over time by layer depth as well as layer type:
>
> https://github.com/anonymousauthor609-coder/icml/blob/main/train_data_threesteps.pdf
>
> https://github.com/anonymousauthor609-coder/icml/blob/main/train_data_type.pdf
>
> Interestingly, we find that EuroLLM seems to follow roughly two learning phases: At first, especially MLPs in the layers close to the outputs seem to drive learning. After step 60k, the influence of intermediate and layers closer to the inputs increases. We will dedicate additional space to the interpretation and discussion of these findings. In the anonymized git above, we also plot the influences of the regularization and first moment, which will appear in the appendix of the revised paper.
>
> > [...]limitations of this method, where does it fail?
>
> To address this, we include a new result where we test the sensitivity of ExPLAIND’s explanation to the efficiency strategies we propose. We find that early accumulation of scores leads no measurable error in the resulting influence scores (i.e. accumulating gradients over batches and then computing dot products is the same as computing dot products of sample-wise gradients and then accumulating scores to the same batch level for 32-bit precision floats).
>
> Second, for subsampling the training steps, we find that the reconstructed prediction of a subsample of steps is close to the actual model prediction in terms of accuracy from around a sparsity of 0.5. In other words, for our MNIST model you can half the number of steps considered and still expect the resulting decomposition to be accurate.
>
> Anonymized figure for all levels of step sparsity: https://github.com/anonymousauthor609-coder/icml/blob/main/mnist_sensitivity_plot.pdf
>
> As we detail above, locally, i.e. output changes observed in single steps, the scores still reproduce exactly. Further, we will elaborate on the above arguments regarding the limitations of this approach, as well as inherent limitations to ExPLAIND compared to other interpretability approaches, such as the lack of causality (line 414), i.e. ExPLAIND does counterfactually predict what would happen, if one were to change any aspect of the training procedure/data. Our method shows what influenced the model instance studied exactly. Compared to other approaches in model component and data attribution, this may not always be what you want: In that case, one should choose a method targeting such form of explanation.
>
>
> > There is currently no comparison of ExPLAIND with other similar methods
>
>
> We agree that comparing against further, non-unified baselines would be very interesting. However, we are not aware of any existing, comparable frameworks that provide explanations that would be directly comparable to ours, i.e. a  model instance-specific, unified decomposition of model outputs into influence scores over data, parameters, and time. We’d be super happy to do such comparisons if you could point us to similar methods we could compare to!
>
> > When discussing the 1.7B model results
>
> It seems there is a part missing from your text in the submitted review. If we couldn’t address the critique related to this above, please let us know!
>
> > I would really like to see the authors open-source the code and hopefully even build a library
>
> We wholeheartedly agree with this and hence already provide our codebase as supplement to this submission. To make ExPLAIND accessible to the community, we of course also plan to publish and maintain a github repository with easy-to-access code for both the small and larger settings.
>
> Thank you again for your consideration. Please let us know if you need any further clarifications on our responses.

---

> > ### Author Rebuttal · Reviewer_is1k · 2026-04-01
> >
> > I thank the reviewers for their rebuttal, this addresses most of my concerns. I especially appreciate the willingness to publish and maintain a GitHub repository implementing their method.
> >
> > What I meant by comparing to other, non unified methods, is for example to compare to existing SOTA data attribution solely on data attribution problems (same for parameter attribution and perhaps pruning). I understand and agree that the unified framework is a strength of this approach, however I am curious to see how it compares in isolated settings.
> >
> > Apologies for the "When discussing the 1.7B model results..." I think what I meant to say is that this sentence "Averaging over the checkpoints, this delta accounts for 0.85 percent of the total loss change." is a bit confusing, it is unclear to me what "this delta" actually refers too and if 0.85% represents the error with respect to the actual loss or if it is a typo meant to say that 85% of the loss can be explained through your method.
> >
> > I am happy to keep my score of 5.

---

> > > ### Author Response · Authors · 2026-04-02
> > >
> > > Thank you very much for your reply and the clarifications.
> > >
> > > > comparing to other, non unified methods,
> > >
> > > Sorry for the misunderstanding! To address the other reviewers' feedback, we have already introduced new experiments comparing ExPLAIND scores accumulated to the data level against two popular baselines, TracIn (Pruthi et al., 2020) and TRAK (Park et al., 2023), as requested. We do this in terms of the linear data modeling score (LDS), the standard metric for evaluating data attribution methods like TRAK through correlation with models retrained on random subsets of the data. We do this on both the CIFAR model and the new MNIST setting. We detail these new experiments in point 1 in our response to Reviewer Hbya. In the first setting, ExPLAIND is on par with TRAK and TracIn. In the latter, we find that ExPLAIND generally performs close to the TracIn baseline, but both are outperformed by TRAK in all but the leave-one-out setting.
> > >
> > > As we discuss in our Limitations section, this is consistent with the intuition that ExPLAIND is not a causal method: Its LDS performance is similarly weak as the baselines. Rather, ExPLAIND attributes the data for a given, single model instance exactly, which is a different objective than that of the baselines, which try to solve the leave-data-out-and-retrain problem (see lines 421 to 428).
> > >
> > > > unclear to me what "this delta" actually refers too and if 0.85% represents the error with respect to the actual loss
> > >
> > > Thank you for raising this. Indeed our wording could have been clearer there. The 0.85% refer to the average error of summing up the scores in comparison to the actual loss change. In other words, our EuroLLM scores reproduce the loss trajectory at the steps considered accurately, only deviating by about 0.85% of the overall change. We will definitely improve our phrasing there in the final revision.
> > >
> > > Thank you again for your feedback.

---

### Official Review · Reviewer_to8v · 2026-03-13

**Soundness:** 3
**Presentation:** 3
**Significance:** 3
**Originality:** 4
**Overall Recommendation:** 5
**Confidence:** 3

**Summary:**

ExPLAIND is trying to answer a question that's always bothered people who work with neural networks: when a model makes a prediction, why did it do that? Not just which input was responsible, but which specific weight, at which training step, trained on which example, contributed how much. The paper builds a mathematical framework that lets you decompose a model's output into these atomic contributions and — importantly — they add up exactly, no approximations. They then use this to study Grokking, which is this weird phenomenon where a model seems to have completely memorized its training data and then suddenly, much later, generalizes perfectly to new examples. What ExPLAIND adds is a precise in-depth account of what's actually happening inside the model during that delayed generalization — which layers activate when, which training examples become influential, and how the internal geometry shifts from a messy memorization solution toward a clean mathematical structure based on cycles and frequencies. The layer replacement experiment is elegant: they extract just two layers from the fully trained model, drop them into a randomly initialized model, and it generalizes almost immediately — proving those two layers are causally doing the work. The only drawback maybe is that none of this works at any meaningful scale.

**Compliance With Llm Reviewing Policy:**

Affirmed.

**Final Justification:**

The limitations have been addressed in the rebuttal/

**Key Questions For Authors:**

1. Theorem 3.1 defines ϕ^train_s with a sum from i=0 to s-1, but the derivation in Appendix equation (6) expands ms+1 with a sum from i=0 to s — the i=s term seems to be missing from the stated theorem. Is this intentional, a typo, or is it silently corrected in the implementation? If the implementation includes it, the theorem should say so. Clarifying this would go a long way toward building confidence in the theoretical foundation.

2. Table 1 validates the EPK reconstruction purely by argmax agreement — but does the full output vector actually match, not just the top predicted class? Even a simple logit-level MSE or max absolute difference between the EPK reconstruction and the original model would make the "exact" equivalence claim much more convincing. Right now this is the biggest gap between what the paper promises and what it actually shows.

3. The 1.7B experiment only reproduces loss curves at batch level — no sample-level influences, no parameter decomposition, which is essentially the whole point of ExPLAIND. How much would it actually cost to run even one sample-level attribution on EuroLLM-1.7B? And is there a realistic path to making this tractable, or is the 1.7B result mostly illustrative?

4. TRAK is the most obvious comparison for data attribution at any meaningful scale and it's not in the paper at all — how does ExPLAIND compare against it on the settings where both can run?

5. The entire flow — the four phases, the alignment finding, the frequency shift — is built on one model doing arithmetic mod 113, which happens to have an unusually clean cyclic structure that almost inevitably leads to a Fourier representation. The obvious question is whether any of this generalizes to other algorithmic tasks that also exhibit Grokking, like sparse parities or permutation groups, which have completely different underlying structure and no reason to produce the same cyclic geometry. As it stands the paper makes implicit claims about Grokking as a general phenomenon but only demonstrates them on a task specifically chosen to have the mathematical properties that make the findings look clean — and those two things are not the same.

**Limitations:**

The paper does include a limitations section, which is appreciated. However it understates how severe the scalability problem actually is — gesturing at it as a challenge without quantifying the gap. Full ExPLAIND on ImageNet-scale models is computationally intractable by several orders of magnitude with current hardware, and the 1.7B experiment gives a somewhat misleading impression that the framework extends to large models when it only reproduces loss curves at batch level — which is not really ExPLAIND. The limitations section would be significantly stronger if it explicitly quantified rather than leaving the reader to work it out themselves.

**Strengths And Weaknesses:**

Strengths

1. The core idea of decomposing a model's output into atomic scores ψs(θ^(i), x, xk)j that add up exactly is mathematically clean and genuinely novel — most attribution methods are approximate, this one not so much.

2. Extending the EPK to AdamW is non-trivial and practically relevant since basically every modern model trains with AdamW, not vanilla gradient descent.

3. The four-phase Grokking story — memorization, circuit formation, alignment, generalization — is a real contribution, and the alignment phase where the embedding and decoder reorganize around the representation pipeline hadn't been identified before.

4. The layer replacement experiment is honest and convincing — taking just the attention and Linear-1 weights, dropping them frozen into a fresh model, and watching it generalize in 200 steps instead of 1939 actually proves something rather than just being suggestive.

5. Figure 4 showing the Fourier frequency shift from ~2 to ~113 as training progresses is really elegant — it's direct visual evidence of regularization killing high-frequency components and selecting the fundamental cyclic representation

6. The framework simultaneously gives you data attribution, parameter attribution, and training step attribution in one unified object — most prior work only does one of these.

Weakness

1. Theorem 3.1 has a likely off-by-one error in the AdamW momentum expansion — the ϕ^train_s definition sums from i=0 to s-1 but the derivation includes the i=s term, which means maybe some gradient contribution just vanishes without explanation — some explanation would be appreciated here!

2. Corollary 3.2 defines b0 = ∇θf(θ0) ≠ 0 but checking step s=1 manually the two formulas don't match unless b0 = 0 — almost certainly a typo but it undermines confidence in the appendix derivations.

3. The validation metric for the EPK reconstruction is just argmax agreement — it only checks whether the top predicted class matches, not whether the actual output vectors agree, which means "100% accurate reconstruction" could be hiding meaningful numerical errors.

4. The scalability situation is somewhat dire — full ExPLAIND on something like ResNet50 on ImageNet would take roughly years of massive compute if I am not wrong, and the 1.7B experiment sidesteps this entirely by only reproducing loss curves at batch level, which is not really ExPLAIND

5. I'm not sure why but there's no comparison against TRAK, which is specifically designed to scale data attribution to ImageNet-sized settings and is the most obvious baseline the paper should be measured against.

6. Everything interesting happens on a single-layer Transformer doing arithmetic mod 113 — one task, one architecture, one phenomenon, and it's unclear whether any of the mechanistic story generalizes to deeper models or more complex tasks.

7. The pruning experiment — the paper's main attempt to show practical utility beyond Grokking — the comparison is against a 2016 baseline — it's hard to read much into; whether ExPLAIND is actually useful by today's standards.

---

> ### Author Rebuttal · Authors · 2026-03-31
>
> Thank you for your thoughtful feedback. We answer as follows:
>
> W1, W2, Q1: Indeed we seem to have introduced a mistake to the step indices of the train feature maps. Furthermore, you are correct about the typo in Corollary 3.2. For the gradient descent with momentum update rule, it should be $\theta_s = \theta_{s-1} - \alpha_s \beta b_{s-1}$.
>
> Our implementations produce the correct decomposition as is verified by Table 1 and there is no impact on our results. We will fix this and do another careful pass. Thank you for checking this!
>
> W3+Q2: We clarify that besides accuracy of the argmax prediction, we also consider the KL-divergence of the full outputs to quantify the agreement of the full output vectors in Table 1. The low KL-divergence in both settings shows that the scores reproduce entire output behaviour, not just the argmax prediction.
>
> W4: You’re right about the complexity of a full ExPLAIND decomposition, which we discuss in Section 4.1 Efficiency and from line 406. However, to gain actionable insights into model behavior, one does not need the entire decomposition: Our experiments show that the influences follow smooth curves, i.e. just looking at a subsampled set of steps would be viable there. Depending on what type of insights you are looking for, materializing only the relevant parts of the tensor of influences  (see lines 237 to 243), for which we provide two examples, is not a limitation for applicability.
>
> To further validate this, we introduce two new settings testing the strategies (early accumulation and step sampling) on our new MNIST model. We find that early score accumulation on the data level has no measurable effect on the scores, while subsampling training steps produces a faithful decomposition until about 50% of the steps. For details, please check the response to Reviewer Hbya, point 2 due to the character limit.
>
> Further, we note that the wall-clock time to compute the ExPLAIND data attribution scores (170s) in the MNIST setting is close to those of TRAK (114s) and TracIn (100s), i.e. in comparable settings, the runtime of ExPLAIND is equivalent to that of other methods.
>
> Q3: For the EuroLLM scores, we in fact use early accumulation only on the data level (to batches) and decompose to the most granular level of single parameters. To address this critique, we re-ran this experiment dividing the batches into sub-batch divisions of the data, each of size 4096 or less tokens, which reflects the size of documents (i.e. single samples). The scores reproduce the loss trajectory in the same way as described in Section 6/Larger Settings, and retains an even finer-grained level of decomposition, now also on the data level. This runs in about the same time as before (about 15 mins per checkpoint; see line 438) with our current implementation.
>
> Furthermore, using the additional page of the camera ready revision, we extend the Section on EuroLLM to also include some qualitative insights which are comparable to the plots shown in Figure 3. (a), i.e. layer attribution over time. Finding similarly interesting patterns (see response to Reviewer is1k, point 1 for details), we argue that this establishes ExPLAIND’s applicability to larger settings. Future work should explore further such directions as well as efficiency of the approach, but we consider such in-depth study out of scope for this paper.
>
> W5+Q4: We address this point by introducing additional evaluations of our vision model as well as a new setting trained on MNIST in terms of the linear datamodeling score (LDS; Ilyas et al., 2022). We find that ExPLAIND scores accumulated to the data level perform comparable to the TracIn and TRAK baselines in the former setting, while in the latter TracIn and ExPLAIND are outperformed by TRAK. For details, please check our response to Reviewer Hbya, point 1 due to the character limit.
>
> W6+Q6: We agree that further study should be dedicated to see whether our findings in the transformer generalize to other settings with Grokking. As we discuss in lines 431-433, we think that such an in-depth study is out of scope for this work as it would require including a much more diverse set of scenarios. Rather, our main goal is to introduce, validate and showcase an interpretability framework. We welcome future work using it as a tool to help us understand generalization more broadly and publish our implementations to enable such studies.
>
> W7: The goal of this experiment is not to establish ExPLAIND as a SOTA pruning method. Rather, we use the baseline to validate that ExPLAIND provides meaningful parameter influences. We chose an established baseline as an orientation for meaningful performance. The pruned model only starts to drop after 90% of the model’s parameters are pruned and still performs nontrivially at only 1%. We think this is an impressive result as it is based on only ranking parameters by our scores, independent of any baseline.
>
> Please let us know if you need any further clarifications.

---

> > ### Author Rebuttal · Reviewer_to8v · 2026-04-03
> >
> > I thank the authors for a responsive rebuttal. The theorem index errors and the Corollary typo I flagged are acknowledged and confirmed not to affect results. The KL-divergence clarification in Table 1 directly addresses my concern about argmax-only validation - the full output distribution is reproduced, not just the top class. The TRAK/TracIn LDS comparison was the experiment I most wanted to see, and the results are honest — ExPLAIND is competitive on CIFAR but trails TRAK on MNIST, which is consistent with its different objective. The EuroLLM re-run at document-level granularity and the layer attribution results meaningfully extend the large-scale story. The scalability concern remains real, but the sensitivity experiments show that step subsampling at 50% still produces faithful decompositions. I am moving my score to 5 — the theoretical novelty, the grokking alignment finding, and the unified framework are solid contributions, and the rebuttal addressed my main concerns substantively.

---

### Official Review · Reviewer_Hbya · 2026-03-13

**Soundness:** 3
**Presentation:** 4
**Significance:** 3
**Originality:** 3
**Overall Recommendation:** 5
**Confidence:** 3

**Summary:**

The paper introduces ExPLAIND, a framework that decomposes model predictions into additive influence scores over the training data, model parameters, and training steps by extending Exact Path Kernels to optimizers such as AdamW. They demonstrate the utility of the framework through a parameter pruning experiment and an analysis of grokking behavior in a Transformer. They also highlight the scalability of the method by proposing techniques to reduce computational cost and demonstrating them on a 1.7B parameter language model.

**Compliance With Llm Reviewing Policy:**

Affirmed.

**Final Justification:**

The authors have answered my questions. I will adjust my score.

**Key Questions For Authors:**

1. How sensitive are the influence scores to the proposed approximations used for scalability?

**Limitations:**

yes

**Strengths And Weaknesses:**

**Strengths**
1. **Well motivated framework:** Understanding the training dynamics of modern networks and attributing predictions across parameters, training data, and training steps can provide valuable insights. The proposed framework offers a principled and elegant perspective for studying these aspects jointly.
2. **Interesting grokking analysis:** The application of ExPLAIND to analyze grokking behavior in the modulo model demonstrates the usefulness and applicability of the method. The findings are interesting and suggest that the approach can provide additional insight into model behaviors and training dynamics, even for a phenomenon that has already been studied extensively.
3. **Clear presentation:** The paper is well written and mostly easy to follow. Limitations of the method, such as the large computational cost, are clearly acknowledged, and the authors discuss possible strategies to mitigate them.

**Weaknesses**
1. **Further empirical comparisons:** While the experiments primarily aim to validate that the influence scores capture meaningful structure rather than to achieve state-of-the-art performance, additional benchmarking against related approaches would strengthen the empirical evaluation and make it easier to understand the practical effectiveness of the framework. Even if direct comparisons to more "specialized" methods such as influence functions are difficult, adding more comparisons would help contextualize the method.
2. **Computational costs and applicability to larger models:** As stated, the authors provide detailed discussions on the computational costs and propose ways to reduce them. It would be interesting to see how sensitive the interpretability results are to their proposed approximations such as subsampling checkpoints and early accumulation of influence scores.
3. The approach has some similarity with the work of
Yolcu et al. (2025): Sparse, Efficient and Explainable Data Attribution with DualXDA, Transactions on Machine Learning Research.
https://openreview.net/pdf/fe8f6f7674be55c912ccb00df97830901cfa5011.pdf
Please carefully discuss the relation.

---

> ### Author Rebuttal · Authors · 2026-03-31
>
> Thank you for your time and encouraging feedback. Regarding the weaknesses and question you raise, we comment as follows:
>
> 1\. We agree that further comparison against other, non-unified approaches would be very interesting, especially for data attribution, where our notion of influence scores, aggregated over training and parameters, allows for a level of comparison. We address this feedback as follows:
>
> Reviewer to8v suggests to compare against TRAK (Park et al., 2023), which is evaluated in in terms of the linear datamodeling score (LDS; Ilyas et al., 2022) which measures the correlation of predictions of new models retrained on a random subset of the data and the predictions as recovered by the influence scores for the respective subsets. We introduce additional evaluations of our CIFAR-2 model and a new model trained on MNIST.
>
> Following the setting in [1], we train a CNN on a subset of 5000 samples of MNIST. To compute LDS, we retrain the model on random subsets of the training data for fractions $\alpha=\{0.5, 0.7, 0.9, 0.95, 0.99, 0.999\}$ (see figure linked below). We compare against single-model TRAK and TracIn (Pruthi et al., 2020) as baselines. Using ExpLAIND, we decompose the outputs of the model and predict the loss of the subset models by computing the respective scores of the respective subset and passing the resulting output through the loss functions. Furthermore, we compute the LDS for the CIFAR model for $\alpha=0.5$, because retraining and attribution is more expensive there.
>
> Anonymized figure at https://github.com/anonymousauthor609-coder/icml/blob/main/mnist_lds_plot.pdf
>
> In the first setting, we find that ExPLAIND generally performs close to the TracIn baseline, but both are outperformed by TRAK in all but the leave-one-out ($\alpha=0.999$) setting. On the CIFAR model, ExPLAIND (LDS=0.06) is on par with TRAK and TracIn (both LDS=0.05).
>
> As we discuss in our Limitations section, this is consistent with the intuition that ExPLAIND is not a causal method: Its LDS performance is similarly weak as the baselines. Rather, ExPLAIND attributes the data for a given, single model instance exactly, which is a different objective than that of the baselines, which try to solve the leave-data-out-and-retrain problem (see lines 421 to 428).
>
> 2\. This is a very good point. We therefore introduce a new experiment showing the sensitivity of our scores to both early score accumulation and subsampling of checkpoints in the new MNIST setting.
>
> Answering your first question, we find that early accumulation of scores leads no measurable error in the resulting influence scores (i.e. accumulating gradients over batches and then computing dot products is the same as computing dot products of sample-wise gradients and then accumulating scores to the same batch level for 32-bit precision floats).
>
> Second, for subsampling the training steps, we find that the reconstructed prediction of a subsample of steps is close to the actual model prediction in terms of accuracy from around a sparsity of 0.5. In other words, for our MNIST model you can half the number of steps considered and still expect the resulting decomposition to be accurate.
>
> Anonymized figure for all levels of step sparsity: https://github.com/anonymousauthor609-coder/icml/blob/main/mnist_sensitivity_plot.pdf
>
> 3\. Thank you for mentioning this very exciting work. We agree that the approach is relevant to our work and hence add the following to our Related Work section:
>
> > Yolcu et al. (2025) propose the DualXDA framework, which combines novel, highly efficient data attribution with input feature attribution to provide explanations unified along data and input features. ExPLAIND further attributes the model parameters and training steps, but does not provide perspectives on the role of the input features.
>
> Please let us know if you need any further clarifications.
>
> **References:**
>
> [1] Deng, J., Li, T.-W., Zhang, S., Liu, S., Pan, Y., Huang, H., Wang, X., Hu, P., Zhang, X., & W., J. (2024). $\texttt{dattri}$: A Library for Efficient Data Attribution. Advances in Neural Information Processing Systems, 37, 136763–136781.

---

> > ### Author Rebuttal · Reviewer_Hbya · 2026-04-08
> >
> > The authors have answered my questions. I will adjust my score.

---

### Decision · Program_Chairs · 2026-04-30

**Decision:**

Accept (regular)

**Comment:**

ExPLAIND proposes a unified interpretability framework that jointly attributes model predictions to training data, model parameters, and optimization steps by extending Exact Path Kernels (EPK) to realistic optimizers like AdamW. The framework is applied to study Grokking in a Transformer, identifying a novel "alignment phase," and is demonstrated on a 1.7B parameter language model.

All four reviewers recommend acceptance, and the consensus is strong. The core theoretical contribution — an exact, additive decomposition of model outputs into atomic influence scores across data, parameters, and training steps — is recognized as mathematically clean and genuinely novel (to8v, Ga8a). The extension to AdamW is non-trivial and practically relevant (to8v). The Grokking analysis, particularly the discovery of the alignment phase and the layer replacement experiment, is an interesting and convincing application (Hbya, to8v). The paper is well-written with clear presentation (Hbya, is1k, Ga8a).

The primary concern across reviewers was scalability: full ExPLAIND is computationally intractable for large models, and the 1.7B experiment only reproduces batch-level loss curves (to8v). The authors addressed this with sensitivity experiments showing step subsampling at 50% still produces faithful decompositions, and extended the EuroLLM case study with document-level and layer-wise attribution. Minor issues include a typo in Corollary 3.2 and an index error in Theorem 3.1, both acknowledged by the authors. The reviewers uniformly found the rebuttal responsive and satisfactory. A technically sound paper with a principled theoretical framework and interesting empirical findings.